

# Chemometric analysis of aerosol mass spectra: exploratory methods to extract and classify anthropogenic aerosol chemotypes

Mikko Äijälä[1], Liine Heikkinen[1], Roman Fröhlich[2], Francesco Canonaco[2], André S.H. Prévôt[2], Heikki Junninen[1], Tuukka Petäjä[1], Markku Kulmala[1], Douglas Worsnop[1,3] and Mikael Ehn[1]

5  1 Department of Physics, University of Helsinki, Helsinki, Finland
2 Laboratory of Atmospheric Chemistry, Paul Scherrer Institute, Villigen, Switzerland
3 Aerodyne Research Inc., Billerica, MA, USA

*correspondence to:* Mikael Ehn (mikael.ehn@helsinki.fi)

**Abstract.** Mass spectrometric measurements commonly yield data on hundreds of variables over thousands of points in time. Refining and synthesising this "raw" data into chemical information necessitates the use of advanced, statistics-based data analytical techniques. In the field of analytical aerosol chemistry, statistical, dimensionality reductive methods have become widespread in the last decade, yet comparable advanced chemometric techniques for data classification and identification

remain marginal. Here we present an example of combining data dimensionality reduction (factorisation), with exploratory classification (clustering), and show the results can not only reproduce and corroborate earlier findings, but also complement and broaden our current perspectives on aerosol chemical classification. We find that applying positive matrix factorisation to extract spectral characteristics of the organic component of air pollution plumes together with an unsupervised clustering algorithm, k-means++, for classification, reproduces classical organic aerosol speciation schemes. In addition to the typical

oxidation level and aerosol source driven aerosol classification we were also able to classify and characterise outlier groups that would likely be disregarded in a more conventional analysis. Evaluating solution quality for the classification also provides means to assess the performance of mass spectral similarity metrics and optimise weighting for mass spectral variables. This both improves algorithm-based classification and provides important clues for a human analyst on the relative importance of variables and data structures.

**1 Introduction**

The research field of mass spectrometry, arguably began already with electromagnetic deflection of ion streams by Wien (1898) and resulting in first veritable mass spectrometers by J. Thomson (1922) and Aston, (1919a; 1919b). The field, now at the ripe age of a century, has since emerged as one the indispensable tools of analytical chemistry (Griffiths, 2008). Advances such as the quadrupole (Paul and Steinwedel, 1953), time-of-flight mass analysers (Stephens, 1946) and electron

ionisation (Bleakney, 1929, 1930), developed during the 20th century have since enabled new applications, such as aerosol mass spectrometers (AMS; Canagaratna et al., 2007b), measuring the composition of particles of the air, with tiniest of mass, online, with high resolving power. The near-universal ionisation method of standardised 70 eV EI allowed the quantitative



measurement of most atmosphere-relevant chemical compounds and their direct comparison with any previous measurements with said ionisation. However, the feat of quantitatively measuring an atmospheric mixture of thousands of known and unknown compounds simultaneously, on a continual basis, produces data in large quantities, presenting a huge challenge for any early adopters of mass spectrometry.

The management, storage and especially analysis and interpretation of this overabundance of information still challenges any analyst of aerosol mass spectrometric results, but fortunately the rapid development of computer capacity and mathematical analysis tools over the recent decades has today allowed the bridging of the gap between experiments and their scientific interpretation, effectively giving rise to the field of chemometrics, i.e. *"using mathematical and statistical methods [...] to provide maximum chemical information by analysing chemical dat*a*"* (Kowalski, 1981; Vandeginste, 1982) in the

1970's. Since then a variety of chemometric applications have emerged in various fields of mass spectrometry, and are common in e.g. biological and medical implementations, food science and chemical engineering (e.g. Belu et al., 2003; Karoui et al., 2010; Kell, 2004; Pierce et al., 2012; Sauer and Kliem, 2010; van der Greef et al., 2004; Wishart, 2007).

Also aerosol mass spectrometry, although a latecomer among mass spectrometric applications, seems to have been quick in adopting and improving on some of the basic tools found useful elsewhere, as exemplified by the surge in use of factor

analytical techniques for data dimensionality reduction in the recent decade (Canonaco et al., 2013; Lanz et al., 2007b; Zhang et al., 2005; Zhang et al., 2011; Marcolli et al., 2006). Yet, there is a considerable amount of work still to be done in the field of aerosol data chemometric analysis – a significant part of more advanced AMS data analysis is still done manually and thus inevitably limited by the expertise and capacity of the human analyst. Especially the classification and interpretation of the AMS spectra are still largely based on a dozen or so mass spectral variables, so called "marker signals", and their

ratios (Aiken et al., 2007; Aiken et al., 2008; Cubison et al., 2011; Farmer et al., 2010; Mohr et al., 2012). Exploring the logical follow up to the automatic data dimensionality reduction (factor analysis): applying similar mathematical, computer-aided tools also for un/semi supervised classification and identification of AMS spectra, has not been performed outside of the specific application of single particle mass spectrometric studies (e.g. Freutel et al., 2013; Liu et al., 2013; Murphy et al., 2003; Rebotier and Prather, 2007). These somewhat underused tools would likely prove invaluable for consistent and

objective analysis of a much wider range of AMS data, lessening the outcomes' dependence on analyst subjective views or reliant on their years of expertise in mass spectral interpretation (Ausloos et al., 1999; Ulbrich et al., 2009). Even for an experienced analyst, exploratory data analysis has the potential to uncover previously unknown underlying mathematical structures within the data (Tukey, 1977), offering invaluable clues for the correct selection of a solutions and their interpretation.

Compared to using only classical factor analytical methods, these more diverse techniques can also uncover the minor, "outlier" aerosol types that often go unnoticed in a long-term factor analysis because of their low relative contribution to total aerosol mass (Ulbrich et al., 2009). Such quantitative information on aerosol chemotypes is widely beneficial for many types of studies involving *e.g.* receptor modelling and aerosol source apportionment (e.g. Canonaco et al., 2013).





In this work we explore the possibility of complementing the current techniques for advanced AMS data analysis with some analytical and processing methods found useful in similar mass spectrometric applications. Namely we will 1) test if a simple unsupervised data clustering method can be used to classify aerosol mass spectral samples without *a priori* information provided by a human analyst, 2) explore data pre-processing to yield optimised spectral similarity metrics and

data pre-processing (Horai et al., 2010; Kim et al., 2012; Stein and Scott, 1994) and thus enhance the structures in data, leading to improved classification (Anderberg, 1973; Spath, 1980). The results on variables' relative importance (optimised weight) for correct grouping of mass spectra can also indirectly help in any manual classification and identification tasks. Finally we will 3) compare which measures of spectral similarity best capture the differences between different atmospheric "aerosol chemotypes" often referred to by the AMS scientific community.

We will exemplify the functionality (and hopefully usefulness) of such a "machine learning" approach, in an analysis of an extensive set of AMS ambient air pollution spectra. We propose the methodology offers in this case an improved way to derive not only reliable, local reference spectra for the archetypical anthropogenic pollution types, but also quantitative estimates for their expected natural variation.

## 2 Methods

Although the focus of this study is in statistical analysis of aerosol mass spectrometric measurements, any such venture is inherently affected by the nature and quality of the experimental data. We will first provide a short overview of the main features, advantages and shortcomings of the aerosol mass spectrometer instrument (Sect. 2.1) and describe the specific set of data we used (Sect. 2.2) in the testing of the methodology – to serve as a background and put into context the choices made when selecting our specific statistical methods and algorithms.

The statistical methods themselves are described in a brief manner in Sect. 2.3, since their in-depth review or commentary is regrettably beyond the scope of this article. We encourage the interested reader to follow the references provided for a more comprehensive account and additional background information on the specifics and inner workings of these data analytical techniques and algorithms.

### 2.1 The AMS instrument and data pre-processing

**2.1.1 Compact time of flight mass spectrometer (C-ToF AMS)**

The data analysed in this study is acquired with an aerosol mass spectrometer featuring a compact time-of-flight (C-ToF) mass analyser. The instrument is developed and manufactured by Aerodyne Research Inc. (Billerica, MA, U.S.). While outclassed mass resolution-wise by subsequent high resolution ToF AMS variants (Canagaratna et al., 2007a; DeCarlo et al., 2006), the C-ToF-AMS does feature a mass analyser superior to many common aerosol mass spectrometers such as the

Aerosol Chemical Speciation Monitor series (Q-ACSM; Ng et al., 2011b; ToF-ACSM; Fröhlich et al., 2013). With highest





sensitivity of any AMS instrument (DeCarlo et al., 2006; Drewnick et al., 2009), the C-ToF-AMS is an advantageous instrument for the purposes of this study: acquiring high precision, unit resolution mass spectra at a high time resolution.

The C-ToF AMS, described in a thorough fashion by Drewnick and co-authors (2005), shares many common characteristics with most ToF AMS instruments – an aerodynamic lens to concentrate the sample aerosol into a tight beam of particles upon entering trough the instruments inlet, a beam chopper to enable particle size measurement based on their flight-time through a vacuum chamber, a thermal vaporiser set at 600 degrees Celsius to flash vaporise the sample, 70 eV electron impact ionisation of the vaporised sample implemented with a tungsten filament, and finally an orthogonal extraction time-of-flight mass analyser to provide the ions' mass spectra.

In the particular C-ToF AMS specimen we used, the particle time-of-flight chamber is considerably shortened (10 cm versus the normal 40 cm chamber). The advantage of this modification is an increased sample transmission from the inlet up to the vaporization and ionization region, at the cost of an increased signal from aerosols' carrier gas, due to the reduction in time and distance for air molecules to diverge from the beam. To combat the effect of increased number of air molecules passing through the system, a helium flow is added after the lens to displace air molecules and increase their molecular diffusion from the beam. With this arrangement the air signal is reduced by a factor of 10 to 100, depending on setting, while only affecting the aerosol signal to a much lesser extent. The He ions are later removed by an additional high pass mass filter located before the analyser. Due to their solitary location at the mass-to-charge ($m/z$) axis, at 4 Th, He molecules and ions are less of a problem than nitrogen and oxygen ions. An additional negative effect of this geometry is also the reduced resolution in particle time-of-flight (PToF) sizing, understandably caused by shortened PToF flight track.

For the above mentioned reasons size segregated data quality is poor for this particular instrument. Fortunately the PToF mode data is nonessential for this specific work, while the excellent signal-to-noise ratio (SNR) available from the standard, non-size-selective "MS mode" clearly benefits any statistical analysis of the data.

### 2.1.2 Pre-processing of AMS data and derivation of final data matrices

In data mining or indeed any statistical analysis, pre-processing and quality assuring the data is an essential step preceding the application of actual analytics algorithms. The AMS data makes no exception. On the contrary — it needs a number of steps and corrections to first compute the per ion mass loading from the raw mass analyser signal, and then to estimate and propagate the errors arising from several sources along the way to the final results. As the general methodology of AMS standard data processing has fortunately been amply described (Allan et al., 2003; Jimenez et al., 2003), as has the derivation of mass spectral matrices and their error estimates (Ulbrich et al., 2009; Zhang et al., 2005), so we will only summarise the procedure here and mostly concentrate on the particulars that deviate from the standard approach in our pre-processing.

The AMS data was automatically corrected for changes in $m/z$ axis calibration, using a time-dependent calibration function and a set of around a dozen known marker peaks to fit the *time-of-flight* to $m/z$ calibration individually to each measured spectrum. The peak areas of all unit $m/z$ signals were then integrated, with manually checked and modified integration regions to yield the background-subtracted signal at each $m/z$ ratio.




The instrument signal response degrade over time was accounted for by using a time period after instrument calibration as a reference point and normalizing the measured air induced signal ("airbeam", $AB$) to the mean value of the reference time ($AB_{ref}$). The normalisation ratio $AB/AB_{ref}$, describing the instrument response to a known mass concentration of air molecules, was used as a scaling factor for all the measured signal, as suggested by Allan and others (2003). Due to concerns

of non-linearity of the detector response with very high signals typical of $N_2$ and $O_2$, argon (Ar) signal observed at 40 Th was used as a metric for the airbeam intensity.

The AMS fragmentation table was slightly modified to accommodate for the issues arising from the modified PToF chamber and the increased airbeam. Namely some air related signal ratios used in the fragmentation calculations, such as relative air contribution to observed signals at 15, 29 and 30 Th, were re-calculated based on exact molecular ratios obtained from filter

runs, automatically performed every three hours. After this re-calibration all the minor, artificial, non-air signals originally seen during filter runs (e.g. in organics and nitrates) at the aforementioned $m/z$ ratios were effectively assigned to the airbeam, excluding them from further analysis.

Efficiency of the ionisation process was determined via ammonium nitrate response factor calibration outlined by Jimenez (2003) and Allan (2003) and co-authors. Finally, collection efficiency (CE; Huffman et al., 2005) was evaluated in relation

to mass derived from (twin) differential mobility particle sizer (DMPS; Aalto et al., 2001) after subtraction of black carbon (BC) given by an aerosol aethalometer (Hansen et al., 1984). The CE correction proposed by Middlebrook and co-authors (2012) was only applied to one ("March 2009") out of the three datasets used in this analysis (see Sect. 2.2), with a small modification to the base CE suggested by the DMPS comparison, as applying it was actually found to slightly weaken the correlation between DMPS and AMS data instead of improving it. For the other two data sets ("May 2008", "September

2008") a constant CE was applied based on a linear least squares fit to best match the mass observed by the AMS against that derived from the DMPS.

Finally matrices consisting of organic mass spectra and their estimated errors at all measurement points, averaged in data acquisition phase to 5 minute intervals, were extracted from the pre-analysis software. The error of the organic signals at each time and for each $m/z$ ratio were estimated and propagated to the output matrices using the standard AMS error

calculation procedure (Allan et al., 2003; Jimenez et al., 2003) within the AMS "Sequential Igor data Retrieval" (SQUIRREL; v.1.50) analysis software programmed in Igor Pro (Wavemetrics Inc, Lake Oswego, OR, USA).

The two matrices, one for organic mass spectra and one for the error estimate were additionally pre-processed using the "PMF Evaluation Tool" (PET; Ulbrich et al., 2009). The pre-processing features mainly take into account an additional correction to the error matrix from electric noise of the instrument, and allow the down-weighting of certain signals with

poor SNR or those derived directly from $m/z$ 44 (thus the variation in $m/z$ 44 carrying too much weight in subsequent data analysis by default). The final mass spectral matrices were then used as an input data for the feature extraction algorithm, explained in Sect. 2.3.1.



## 2.2 Site, measurement campaigns and identification of air pollution events

The experimental data used in this statistical analysis exercise originate from long-term ambient air observations at the SMEAR II station in Hyytiälä, Juupajoki, Finland. The particulars of the measurement campaigns, environment and pollution events are described below.

### 2.2.1 SMEAR II field site in Hyytiälä and the EUCAARI measurements

As the practical example of the study is about applying our exploratory data analytical techniques to resolve the archetypical air pollution classes typical of a non-urban field site, we obviously strive for a high quality, variable and comprehensive set of data to refine and test the methodology. Our datasets of choice originate from the comprehensively documented and characterised station of SMEAR II in Hyytiälä, during the well-covered intensive measurement EUCAARI campaigns of 2008 to 2009 (Kulmala et al., 2009). This already rather familiar data will offer a good testbed for the proposed methodology. Below both the station and the intensive measurement campaigns are described.

Situated in Southern Finland (61°50'40"N 24°17'13"E) amidst subarctic pine forest, the Hyytiälä forestry station and the collocated "Station for Measuring Ecosystem–Atmosphere Relations" (SMEAR II; Hari and Kulmala, 2005) offer an environment well representative of the vast taiga biome of Northern Eurasia. While not exactly pristine, the surrounding lands are rather unbroken homogenous production forests consisting mainly of typical Scandinavian and Russian taiga tree species; pines (*Pinus Sylvestris*) spruce (*Picea Abies*), and to a lesser extent birch (*Betula Pendula, Betula Pubescens)* and other deciduous broadleaf species (e.g. species from *Populus*, *Alnus* and *Sorbus* genera). Williams and co-authors (2011) estimate from land use statistics 94% of the local (5 km radius) and 90% in the nearest 50 km land area consist of forested land (including forest at seedling or sapling state). The nearest town of Orivesi (pop. 9500) lies 19 km due South of the station and the city of Tampere (pop. 213 000) c.a. 48 km to South-West. The surrounding county of Juupajoki is sparsely populated (5–10 inhabitants per km$^2$) and while it does have some local sources anthropogenic air pollution, such as household heating and cooking, they are generally very limited in terms of magnitude and geographical breadth.

A notable exception to the absence of major anthropogenic pollution sources in the local environment are the two lumber mills and the wood pellet factory in the small village of Korkeakoski, 7 km East-South-East from Hyytiälä, and the other small sawmills further away, which do have a marked influence on the volatile organic compounds (VOC) concentrations and aerosol population of Hyytiälä when the incoming airmass is advected over these mills (Eerdekens et al., 2009; Liao et al., 2011). Additionally, although the local aerosol sources such as passing vehicles, cooking emissions at the forestry station, nearby cottage, household, or sauna heating are negligible in large scale, they can momentarily affect the local air quality if emitted from close enough not to be diluted below the detection limits, and are thus detected by many of the SMEAR II stations more sensitive, high time-resolution instruments.

Nevertheless, the preeminent cause of degraded air quality (relative to the background) at the station is the medium-to-long range air convection from industrialised areas of Southern Finland and especially the St. Petersburg region in Russia



(Kulmala et al., 2000; Patokoski et al., 2015; Riuttanen et al., 2013) and even all the way from the industrial heartlands of (mostly Eastern) continental Europe (Niemi et al., 2009; Sogacheva et al., 2005).

Independent of these anthropogenic components, the local atmosphere is always influenced by the ever present biogenic background aerosol and biogenic volatile organic compounds (BVOC). These exhibit their own seasonal and diurnal variations; BVOC's concentrations are generally high both during afternoon due to emissions maximum and nighttime due trapping of emissions in the shallow, unmixed boundary layer (Rinne et al., 2005) and the aerosol biogenic particle mass also due to thermally driven condensation of semi-volatile species into existing seed particles. Due to the biological origin of the natural aerosol, the biogenic aerosol background is obviously higher in warmer months (e.g. Patokoski et al., 2014).

The relative lack of local anthropogenic sources and their pronounced dependence on airmass origins, manifesting as observations of isolated aerosol and gas phase plumes, make Hyytiälä an ideal natural laboratory for studying the effects and characteristics of local and transported air pollution on the otherwise clean atmosphere over the expansive subarctic biomes.

The EUCAARI study, conducted in years 2007 to 2010 aimed at examining the interactions between air pollution and climate change (Kulmala et al., 2009). The results have been widely published (see Kulmala et al., 2011, for a summary of findings), and include discussion on e.g. aerosol source apportionment and chemical ageing (Kulmala et al., 2011; Ng et al., 2010b). The intensive observation periods of the project took place in spring 2008, autumn 2008 and late winter 2009. The exact timeframes of the AMS measurements are available in Sect. 3.1, Table 1.

An especially comprehensive analysis of the EUCAARI intensives' AMS data is written by Crippa and co-authors (2014), providing us a reference point to compare our results with. Their analysis provides plausible estimates of aerosol speciation using consistent methodology, but due to the obvious limitations induced by the very large number of datasets and the somewhat rigid methodology of using general reference spectra from quite different types of environments for all the sites, and applying rather strict constraints to their allowed variability, it is possible for the analysis to miss out on some divergent, locally relevant phenomena. As an additional motive for this work, we aim to provide considerably enhanced prerequisite information for applying such a factor analytical methodology for an individual site, by observing the local anthropogenic aerosol characteristics and variation, and tailoring the input reference spectra and variation estimates accordingly.

### 2.2.2 Identification and selection of air pollution events

The term "air pollution event" is in the context of this study defined loosely as a period of significantly increased concentrations relative to a stable background aerosol. This implies the pollution episode has a distinct beginning, a point in time when a relatively stable background aerosol is first complemented with a specific pollution aerosol, and an end, when the pollution vanishes, leaving a background aerosol similar in composition and mass to that observed before the event. During the pollution episode we assume the observed total aerosol is a two (or in some cases multi) component superposition of background and a chemically invariable "pollution plume" aerosols. Although there undoubtedly are aerosol dynamical processes ongoing between the two aerosol types, the background and the plume, we assume these to be of minor importance



due to the generally short timeframes of the events. Some examples of typical pollution event types and durations are given in the results (Fig. 1).

When basing the pollution event definition on aerosol mass, while we can with confidence pick out the clearest instances of pollution, we run into problems when the increase in mass concentration approaches the magnitude of noise in the instrument. Also after long pollution events, lasting a better part of a day or longer, it is questionable if the background aerosol has remained the same and, if it is even possible to describe the changing pollution unambiguously. To address these issues of demarcation, we needed to define further conditions to separate real pollution from on one hand short instrument noise peaks and on the other, long periods of increased concentrations of gradually changing or evolving pollution aerosol. Hence the requirements of a pollution episode to be accepted for our analysis were set as follows:

1. Temporary, distinct rise of organic aerosol mass concentration above background level.

2. Being able to unambiguously separate the pollution plume from the background.

3. The supposed pollution spectra needs to be physically reasonable. (i.e. not a minor spike of instrument noise).

It should also be noted, that while the first and third conditions could be examined using a simpler, non-statistical analysis method, like background subtraction, there are many pollution episodes where this would not suffice. While we could define the "pollution plus background" spectrum from the event period and the "background" spectrum as an average of background before and after the event, and subtract the latter from the previous, yielding the characteristic "pollution" spectrum; applicable e.g. in Fig. 1 (panel a), the second condition is much more problematic if the background varies, not even necessarily in a linear way, or several pollution plumes overlap (e.g. the case shown Fig. 1, panel d). This applies especially to events with long duration, from several hours to even a couple of days, and to events with multiple consecutive peaks, which still represent the same event (case in Fig. 1, panel b). To account for these complications we feel a more advanced data reduction method, such as the factor analytical approach presented in Sect. 2.3.1, is indeed required to be able to thoroughly evaluate events' satisfaction our selection criteria.

## 2.3 Statistical analysis tools

As numerous studies have already been conducted on feature extraction and data dimensionality reduction in connection with AMS results (Ng et al., 2010a; Zhang et al., 2011), in this work we will focus more on classification and identification techniques suitable for AMS data.

Based on a brief review of suitable data analytical methods, we selected the specific methods and algorithms used in this work: for pollution feature extraction we selected a model already tried and tested for AMS data, the Positive Matrix Factorisation (PMF; applying the ME-2 algorithm; Paatero, 1999) and for feature classification we use the elegantly simple and long-established k-means clustering algorithm (MacQueen, 1967). While any selection of a method inevitably opens the door for various arguments on using a particular model, algorithm or parameter over another, we wish to emphasise the purpose of the study is not to find the ultimate or optimum method or to offer final answers, but rather serve as a first



approximation, an opener for the development of more refined and better optimised classification methodologies and a baseline against which to compare more advanced techniques.

In this work we will use PMF in a non-classical way, to (nearly) unambiguously extract characteristic air pollution spectra from air pollution events. As there often exists considerable variation among mathematically equally good PMF solutions,

termed "rotational ambiguity" (see supplementary material Sect. S.1), the issue of selecting the correct solution needs to be resolved. We propose that in the context of this work, selecting the PMF solution with non-correlating time series of plume and background can be used to identify the rotation that best separates the characteristic pollution factor, and thus to largely avoid the PMF's Achilles heel of rotational ambiguity. We will then demonstrate classification of the extracted spectra to aerosol types, using k-means clustering, and study the effects of simple data pre-processing options and basic metrics for

spectral (dis)similarity on these classification solutions.

### 2.3.1 Positive matrix factorisation (PMF) and its application to studying air pollution plumes

In the analysis of aerosol mass spectrometric results, data reductive methods are put to good use for reasons explained in the introduction. To respond to the challenges and requirements posed by the AMS data, experts in statistics and modelling have updated many traditional analysis tools and developed new ones to answer the specific needs of this type of environmental

data analysis. Perhaps the best known technique developed specifically for feature extraction from environmental, multidimensional data is the use of the positive matrix factorisation (PMF) model to de-convolve and interpret the enigmatic organic aerosol chemistry reflected by the often complex AMS mass spectra.

The PMF technique developed by Paatero and Tapper (Paatero, 1997; Paatero and Tapper, 1993, 1994) is an iterative, factor analytical model to explain observations at a receptor site, i.e. time series ($t$) of variables ($v$) (in form of a size $t{\times}v$ matrix

"$X$"), using a bilinear combination of temporal behaviour of loadings of factors ($f$ in a $t{\times}f$ matrix "$G$") and the factors' time-invariant profiles (an $f{\times}v$ matrix "$F$"), describing composition. If then $E$ denotes the unexplained residual, the difference between the model ($G{\cdot}F$) and the observations ($X$), the PMF model can be formulated

$$X_{(t{\times}v)} = G_{(t{\times}f)} \cdot F_{(f{\times}v)} + E_{(t{\times}v)} \tag{1}$$

where the subscripts indicate the sizes of matrices, corresponding to the number of points in time series ($t$), number of factors

($f$) and number of variables ($v$). While t and v are decided by the set of data available (and possible pre-processing), $f$ is essentially a free parameter selected by the analyst, as is apparent from Eq. (1). Importantly, in PMF all the entries in each of these matrices are limited to positive values, corresponding to environmentally relevant loadings and profiles being non-negative. This considerably reduces the amount and variety of mathematical solutions to the modelling problem, and helps to effectively filter out some of the physically unrealistic, negative solutions (Paatero, 1997).

One of the noteworthy improvements, summarised in detail by Paatero and Tapper (1994), over previous feature extraction methods such as principal component analysis (PCA; Hotelling, 1933; Jolliffe, 1986; Pearson, 1901) is, the PMF model does not blindly minimise $E$, but rather the *weighted* residual. This allows for better measure of the amount of variation not





explained by noise from the experimental measurement, denoted by the standard deviation of variables, (a size $t \times v$ matrix "$\sigma$").

Therefore the objective function to be minimised, "$Q$", can be written

$$Q_{t \times v} = \sum_{i=1}^{t} \sum_{j=1}^{v} \left(\frac{E_{i,j}}{\sigma_{i,j}}\right)^2 \qquad (2)$$

i.e. the squared residual to be minimised is effectively scaled by the variance of each point in the matrix.

In this work we utilise PMF in a non-standard way, to resolve the time series and mass spectral profiles explaining "anomalous" observations often discarded from a PMF analysis, the periods with air pollution spikes and plumes. The PMF analysis is done for each air pollution event (defined in Sect. 2.2.2) individually, altering the time window of the PMF analysis around the event to include both the pollution episode and some background before and after the event. The

advantage of studying this type of relatively short term phenomena is that we can easily evaluate fulfilment of the criteria outlined in Sect. 2.2.2, and we can additionally discriminate between mathematically equal solutions, mostly evading the issue of rotational ambiguity. Essentially knowing beforehand what the (qualitative) temporal behaviour of a pollution and background factors should be like, (*i.e.* the time series of the factors should be uncorrelated), we explore the number of factors and the solution space to select the solution best fulfilling our criteria for a physically correct solution. Adhering to

these criteria, we strive to minimise the ambiguity related to our selection of solutions, as well as considerably reduce the effect of subjectivity with regard to selection of solutions.

### 2.3.2 The k-means algorithm

K-means clustering is one of the most popular, widely used and well known classification algorithm developed already back in 1950's and 60's (Ball and Hall, 1965; MacQueen, 1967; Steinhaus, 1956). It is a simple, iterative, partitioning clustering

algorithm that partitions a set of objects in multi-dimensional space into pre-set number ($k$) of clusters based on a distance (or dissimilarity) metric. For each cluster resulting from any partitioning solution we can calculate a quantity measuring the cluster's "cohesion", a within cluster sum of squared distance between the calculated cluster centre $\mu_n$ (of a cluster $c_n$) and all member objects $x_i$ assigned to it. In Euclidean space we get:

$$J(C_n) = \sum_{x \in C_n} \|x_i - \mu_n\|^2. \qquad (3)$$

The k-means algorithm tries to minimise this quantity $J(C_n)$ summed over all clusters $k$, which we denote $J(C)$:

$$J(C) = \left(\sum_{n=1}^{k} J(C_n)\right). \qquad (4)$$

The iterative procedure of k-means is briefly described in supplementary information (Sect. S.2). Upon convergence on a solution (i.e. global or local minimum of $J(C_n)$) the output of the algorithm gives the user the final assignment of points to clusters, the cluster centroid locations $c$ as well as distances from each data point to all other points and the cluster centres.

These distances can be used to evaluate the quality of both the entire clustering solution and the cohesion and variance of individual clusters. It is important to note k-means converges on any minimum of Eq. (4) found, regardless of if it's global or





local. Finding the global minimum is not guaranteed but can be made more probable by performing repetitive clustering with different initialisation for starting cluster centres and selecting the result with lowest $J(C)$.

Further discussion on selection of user parameters for k-means initialisation and cluster numbers is presented in Sect. 3.2 and in the supporting material (Sect. S.2). For this analysis we used k-means algorithm applying an improved initialisation

method (kmeans++; Arthur and Vassilvitskii, 2007), and the number of clusters ($k$) was kept as a free parameter within a range of $k = 2$ to 20. The selection of dissimilarity metric parameter is discussed below.

### 2.3.3 How to define (dis)similarity of mass spectra?

Among the most important questions in clustering is the selection of measure for "distance" or "(dis)similarity" between two objects, a topic where there are both theoretical (Anderberg, 1973) and experimental (*e.g.* Stein and Scott, 1994)

considerations to be taken into account. Fortunately for the choice of metric we have plenty of recommendations available for our selection: there are several guidelines and recommendations (e.g. Cormack, 1971; Gordon, 1999; Kaufman and Rousseeuw, 2009) available of which similarity metric best to apply for various types of problems, including problems related of identification, comparison and classification of mass spectra similar to ours. As an experimental basis for the metric comparison we cite the informative and thorough study by Stein and Scott (1994) of NIST Mass Spectrometry Data

Center, the conclusions of whose are covered in wider detail further below. Importantly, the distance metric selected needs to be mathematically compatible with the type of variable on hand. This point in question is addressed in the supporting material Sect. S.3.

Some common approaches available for and often used as distance ($d$) metrics include:

1) The squared Euclidian "distance":

$$d(u, v) = \sum_{i=1}^{n} \|u - v_n\|^2 \tag{5}$$

2) The cityblock distance (or "Manhattan distance"; Johnson and Wall, 1969; Carmichael and Sneath, 1969)
$$d(u, v) = \sum_{i=1}^{n} \|u - v_n\| \tag{6}$$

3) Cosine "distance" (Sokal and Sneath, 1963):
$$d(u, v) = 1 - \frac{u \cdot v}{\|u\| \|v\|} \tag{7}$$

4) Correlation "distance" (Fortier and Solomon, 1966; McQuitty, 1966; Sokal, 1958)
$$d(u, v) = 1 - \frac{\sum_{i=1}^{n}(u_i - \bar{u})(v_i - \bar{v})}{\sqrt{(\sum_{i=1}^{n}(u_i - \bar{u})^2)}\sqrt{(\sum_{i=1}^{n}(v_i - \bar{v})^2)}}, \tag{8}$$

where $u$ and $v$ are $n$ dimensional vectors corresponding to objects (with the subscript $n$ here corresponding to the *m/z* variables), and $\bar{u}$ and $\bar{v}$ respectively the mean of variables in $u$ and $v$. Although often called "distances", the squared Euclidean, cosine and correlation measures are strictly speaking not "distance metrics", as they violate the triangle equality

required of a proper distance metric, and should be considered instead a measures (metrics) of dissimilarity between a pair of objects (Anderberg, 1973; Spath, 1980). Other metrics obviously exist as well, but as a comprehensive review is





unfortunately out of the scope of this work, we limited our comparison to these common metrics available in our analysis software (Matlab 2015a, MathWorks Inc., Natick, MA) standard functionality.

Additionally to experimentally evaluating the metrics, Stein and Scott (1994) recommend data weighting methods such as signal intensity scaling and mass scaling to be examined. They find modest improvement of a couple of percent in accuracy

in the dot product (cosine) and Euclidean based matching, when scaling the signal intensities by their square root to emphasise smaller signals, or when scaling all the signals by a power of their "mass" (i.e. $m/z$ ratio), placing more weight on the higher $m/z$ signals as a pre-processing measure. For intensity scaling the weight given to a variable (signal at $m/z$) can be expressed as

$$\mathrm{weight_{intensity}} = \sqrt[s_i]{\mathrm{signal}}, \tag{9}$$

where $s_i$ is a (root function) intensity scaling factor. For mass scaling the variable weights are given by

$$\mathrm{weight_{mass}} = \left(\frac{\mathbf{m}}{\mathbf{z}}\right)^{\mathbf{s_m}}, \tag{10}$$

where $s_m$ ( $> 1$) is a mass scaling factor for the variable locations ($m/z$). We also test these in connection to our data and report the results in Sect.3.2.3.

Although the theory and literature seem to favour the cosine (dis)similarity as a measure of mass spectral objects' association

to each other, we ran several comparisons using different parameters for k-means++, and present the results in Sect. 3.2. To objectively evaluate and interpret the classification results, we additionally pursued a metric, other than "expert opinion", for measuring the "quality of a solution". Some alternative evaluation options are discussed and our method of choice, the silhouette examination, is presented below.

### 2.3.4 Silhouettes in evaluation and interpretation of clustering solutions

To evaluate, and to an extent validate, the clustering analysis we need an objective, diagnostic metric for comparison of different results. There are several alternatives available, four of which we tested in relation to this work. We considered the four evaluation criteria available in the Matlab software statistics toolbox (R2015a), namely the "silhouette". (Rousseeuw, 1987), "Calinski–Harabasz" (Caliński and Harabasz, 1974), "Davies–Bouldin" (Davies and Bouldin, 1979) and "gap" (Tibshirani et al., 2001) criteria, the quick evaluation results of which are presented in supporting information S.4, Fig. S.2 to

S.4).

The downside of the three latter methods tested is that they do not (at least unmodified) accept all non-distance dissimilarity metrics such as the cosine (dis)similarity. For squared Euclidean distance, which was compatible with the evaluation functions, the methods yield mixed results. Upon examining the k-means solutions as described below in the results section, as well as based on theoretical considerations (Sect. 2.3.3, S.3) we feel the use of non-Euclidean metric may indeed be

recommendable, and that the silhouette criterion does manage to convincingly identify the number of "natural", physically reasonable aerosol types (clusters) – therefore we will opt for using the silhouette value criteria detailed by Rousseeuw (1987) in our evaluation of the clustering results of this work.





Rousseeuw (1987) defines for each object *i,* belonging to cluster A and having B as the nearest neighbouring cluster, a silhouette value of *s(i)*:

$$ s(i) = \begin{cases} 1 - \frac{a(i)}{b(i)}; \; for \; a(i) < b(i) \\ 0; \; for \; a(i) = b(i) \\ \frac{b(i)}{a(i)} - 1 \; for \; a(i) > b(i) \end{cases}, \tag{11} $$

where $a(i)$ is the average distance to all other objects of the same cluster (A), and $b(i)$ is the average distance to all objects of

the closest neighbouring cluster (B). For a singleton cluster, containing only one object, $a(i)$ is not well defined. Rousseeuw puts $s(i)$ to zero in this case, but other conventions exist that use a silhouette of one for singletons.

The silhouette value has some convenient properties for interpreting the quality of the clustering assignments that can be applied on single point, cluster and total solution levels. When $s(i)$ is close to unity, the within cluster dissimilarity $a(i)$ is much smaller than the between cluster dissimilarity $b(i)$, indicating the point is very likely correctly grouped, and conversely,

classifying the point the next nearest cluster would be a much poorer choice. On the other hand if $s(i)$ is close to -1, it signifies the next best clustering choice would actually be a much better one than the current assignment, i.e. the point is on average more similar to the points in the neighbouring cluster than to the points in its assigned cluster. This implies the point is likely misclassified. If $s(i)$ is close to zero, the point is situated between clusters, and it is not at all clear to which it belongs to – its dissimilarity to both of the groups is about equal ($a(i) \approx b(i)$).

The average $s(i)$ of points in a cluster, *average silhouette width*, expresses if a cluster is clear cut or weak: the higher the average cluster silhouette width, the more pronounced the cluster is. A graphical representation displayed in supplementary information (Fig. S.5). The *overall silhouette width* is the average $s(i)$ of all the objects, and can be used a parameter to judge the overall quality of the clustering solution. Maximising overall silhouette value can be used to evaluate the "natural" number of clusters in the data (Rousseeuw, 1987), an approach we will also utilise in this work. Some further notes on

silhouette values can be found in supplementary material (Sect. S.5).

### 2.3.5 Posterior processing – weighting of cluster centres and deriving within cluster variation

The k means algorithm yields a list of assignations of all objects to clusters, and also provides the cluster centres defined as the arithmetic mean of the objects within the cluster. These artificially constructed centres can be used to denote the average object within that cluster. However, this approach is subject one of the main weaknesses of k-means, the susceptibility to

outliers, borderline cases and outright mis-classifications also affecting the cluster centre location equally as the objects that would be considered very appropriately clustered. These derive directly from the simplistic functionality of the k-means algorithm (Sect.2.3.2), and therefore there is little to be done to alleviate the issues outside of selecting another algorithm (with its own unavoidable weaknesses).

Nevertheless, we do have additional, diagnostic information available to us outside of the simple list returned by the k-means

algorithm; in form of the silhouette value information calculated from the assignation listing combined with the dissimilarity





matrix. In this work we aim to utilise the statistical information available to us to the fullest, and in the spirit of this goal we apply a simple post-processing step to derive weighted centroid objects to represent the groups of objects in a more robust, classification error resistant way.

Obviously the objects nearer the cluster centre are a better representation of the class than the ones on the edges, or indeed

the ones likely misclassified; ergo they should carry more weight when a typical representative of the class is selected or constructed. In this work we construct "characteristic" centroid objects, *i.e.* spectra, by taking a weighted arithmetic mean of the cluster members instead of the original, unweighted sample mean. As weight we use the silhouette values indicating the confidence we have on the "representability" of the object. Any likely misclassified objects with negative silhouette values have their weight set to zero. The weighted cluster centroid $C_w$ can be expressed as

$C_w = \frac{\sum_{i=1}^{n} u_i w_i}{\sum_{i=1}^{n} w_i}; \quad w_i = \max(0, s(i)),$              (12)

where $u_i$ are the cluster member objects and $w_i$ the respective weights, *i.e.* the non-negative silhouette values *s(i)* obtained from Eq. (11). Similarly we obtain a weighted standard deviation $\sigma_w$ for a measure of the within-cluster variation. With Bessel correction (Gauss, 1823) for small samples' variance we can write for the weighted standard deviation:

         $\sigma_w = \sqrt{\frac{\sum_{i=1}^{n} w_i (u_i - C_w)}{\sum_{i=1}^{n} w_i - \frac{\sum_{i=1}^{n} w_i^2}{\sum_{i=1}^{n} w_i}}}.$             (13)

The change in mass spectrum induced by the weighting was determined to be extremely low, as can be seen comparing the unweighted and weighted spectra, exemplified in supporting material, Fig. S.6. For the final spectral solution presented in this work, the similarity ($r_s^2$, [Pearson] coefficient of determination for mass-scaled spectra) between the scaled and unscaled centroids was found to range from 0.994 to 1.000), confirming that weighting by silhouette does not markedly alter the resulting spectra.

Overall the variabilities represented by the weighted standard deviations are generally smaller than the unweighted ones, due to the down-weighting of outlier objects' influence. By this post-processing we hope to derive more representative "characteristic mass spectra" for the pollution types, and to decrease the error from the ambiguity of the classification. This, we hope, allows us to instead derive plausible estimates for the actual *natural variation* within a specific aerosol type.

## 3 Results & Discussion

In the following chapter, we present some examples of pollution spectrum extraction (Sect. 3.1), and evaluate the similarity and weighting parameters used for their subsequent grouping (Sect. 3.2). We then offer an aerosol chemical interpretation for the different aerosol types (clusters) for the grouping we consider most realistic (Sect. 3.3 and 3.4) and further try to understand and interpret the meta-structure of clustering solutions, i.e. how the solutions relate to each other, what drives them, and how they are related to divisions in chemical characteristics (Sect. 3.5). Finally some basic estimates of natural

variability within the pollution types are given in Sect. 3.6.



### 3.1 Extraction of pollution spectra

Although time consuming, applying the pollution feature extraction approach (described in Sect. 2.3.1) to the identified pollution events (Sect. 2.2.2) allowed us to extract the pollution factors' spectral profiles. Applying our simplistic selection criteria to find the physically most correct rotation among the solutions, we hope to have minimised the rotational ambiguity,

as well as the need for subjective choices by the analyst. Following the procedure described in Methods, we managed to extract a total of 81 characteristic mass spectra, corresponding to as many unique pollution plumes. Some supporting information, namely the local time and above canopy wind direction taken at the time of peak mass concentration was recorded for all plumes. The background spectra were not further considered in this analysis. The per-campaign distribution of the successfully extracted pollution events as are presented in Table 1.

Some examples of factor time series of various types of accepted extractions are given in Fig. 1, illustrating the considerable (temporal) variability among the types and conditions of pollution events, e.g. from a single plume with stable background, (Panel 1a); to very complex event with multiple overlapping plumes (Panels 1d and 1e).

### 3.2 Evaluation of clustering parameters and pre-processing options

As discussed in Sect. 2.3.2, there are several options for the standard k-means clustering, particularly in terms of data pre-

processing, selecting the number of clusters and the distance metric, but also in specifying the number of repetitions, type of clustering initialisation and treatment of "empty clusters' during the iteration process. In the course of data analysis, we explored the effects of these parameters and pre-processing options on the quality of our clustering solutions and their general structures.

### 3.2.1 General clustering parameters

We note that using a low repetitions number ($< 10$) does not reliably return the exact same, optimal solution, so there seem to be several similar, but non-identical, local minima different from the global one, for k-means to convergence on. A hundred or a thousand repetitions already seem to offer consistent and reproducible results. In evaluating the effects of pre-processing, a thousand repetitions were used and in calculation of the results selected for detailed chemical evaluation (Sect. 3.4), the algorithm was run ten thousand times.

Clustering initialisation method was not found to notably affect our results in any way, at least with generally high number of repetitions used. Due to literature recommendation based on comparison (Shindler, 2008) the "k-means++" initialisation by Arthur and Vassilvitskii (2007) was thus selected for use.

We set the "empty cluster action" additional option, i.e. what happened if an empty cluster is created in the course of the iterative process, as "singleton", meaning the point with highest distance score to its cluster centre was assigned as its own

cluster. This forces the solution to always conform to the original cluster number. Generally, an empty cluster was produced





in much less than 1% of all the total iterative processes, so we do not consider this to have affected the overall result, especially since $k$ dependence of solution quality was in any case also studied.

The selection of cluster number $k$ is unquestionably of high importance, as is the selection of dissimilarity metric (Anderberg, 1973; Spath, 1980; Hastie et al., 2005), so they were more thoroughly and quantitatively investigated. Since the

5    above mentioned parameters were generally found to have a major effect on the clustering outcomes, they were not fixed, but kept as free parameters throughout the rest of the testing phase. This allowed us to observe if the pre-processing procedures' effects would be $k$ or dissimilarity metric dependent. The results of applying the commonly used pre-processing options, namely the intensity and mass scaling procedures recommended by among others Stein and Scott (1994), Horai and co-workers (2010), are presented below.

### 3.2.2 Solution quality without pre-processing

Having no definitive preconception on the number of clusters, we evaluated clustering results for a range of $k$'s. ($k = 2…20$) for all the metrics studied (squared Euclidean ["sqEucl" or "Euclidean"], cityblock / Manhattan ["city"], cosine ["cos"], correlation ["corr"]). Using total solution silhouette value as a solution quality indicator, we search for the maxima (or clear elbows) in the silhouette results (Fig. 2), implying particularly favourable solutions.

Based on the silhouette value comparison for the unscaled data (Fig. 2) we conclude the following: the cityblock distance metric seems to perform poorly compared to the other three alternatives. The squared Euclidean, correlation and cosine methods are more or less equal in their silhouette quality, making the selection based on this test alone a difficult task. We also find the silhouette values for the latter methods between values of 0.25 and 0.50, suggested by Kaufman and Rousseeuw (1989) as a region of "weak structure" in the set of data, and calling for use of additional methods to probe if the implied

structure is real or artificial. We additionally note there is clear variation in silhouette values as a function of $k$, indicating lower range ($k < 11$) solutions are more likely to correspond to natural divisions in the data than the high range ($k > 11$). In the following tests we therefore decided to include the range of $k = 2$ to $k = 12$.

For additional visualisation, similar diagnostics for when (not metric specifically optimised) mass scaling is applied are also presented in Fig. 2. The example mass scaling factor $s_m$ of 1.36, selected for the briefly illustrating the effect from scaling,

was selected based on a more comprehensivereview presented below.

### 3.2.3 Solution quality with mass and intensity scaling

As pre-processing options we also tested the two methods recommended by Stein and Scott (1994), namely intensity and mass scaling of the data variables, as explained in Sect. 2.3.3, Eq. (10). Similar to Stein & Scott, we also explore values for $s_i$ ranging from $1…2$ and $s_m = 0…3$, with a step of 0.01 to pinpoint any maxima and evaluate the stability of the results with

regard to minor changes in scaling values. The resulting 2-d field of solution silhouette values is shown in Fig. 3, and can be thought of as an extension to Fig. 2, which corresponds to the situation for scaling factors $s_m = 0$ and $s_i = 1$. Generally, the mass scaling processing was found to enhance the cluster-like structuring of the data, enabling improved differentiation





between groups. It seems also there is no single value of $s_m$ that would maximise the structure, but the optimum scaling factor value depends on number of clusters ($k$). Even so, $s_m$ values between 1 and 2 seem to produce the highest silhouette values for all metrics. If opting for the use of a single $s_m$ value for similar AMS data, we therefore suggest based on $s_m$ distribution of solutions shown in Table 2, an $s_m$ of 1.36 ± 0.24 (mean ± stdev) to be examined as a starting point. When

comparing the scaled result silhouettes from non-pre-processed data, the improvement is non-homogeneous, and seems to specifically enhance some solutions over others, as illustrated in supporting material Fig. S.7.

Similarly, intensity scaling was charted for $k = 2…12$ and $s_i = 0…3$. However, unlike mass scaling, intensity scaling only seems to deteriorate solution quality for our AMS set of data, for the entire range of $s_i$ values tested (0 to 1). Effect of intensity scaling is illustrated in supporting material Fig. S.8 and S.9. Based on this result, we would not recommend

intensity scaling for a data set of this type without further results to the contrary.

Finally, we tested the combination of mass and intensity scaling, but found the results worse than for mass scaling alone. We additionally tested two methods with similar aims, namely omitting the low end mass spectrum < 45 Th and down-weighting $m/z$ 44 related signals, similarly to the standard procedure in PMF matrices pre-processing explained in Sect. 2.1.2. While omission of low masses seems to generally improve classification considerably (supplementary information Sect. S.3. and

15 Fig. S.1), we find the method too arbitrary to recommend, and find mass scaling can be used to produce similar results with better founded, more elegant methodology. The tests on $m/z$ 44 down-weighting were inconclusive at best, and would require further testing to be validated as a procedure with positive effects on clustering structure.

In conclusion of the pre-processing methods, we find mass scaling is the only method to consistently (but non-homogeneously) enhance the data cluster structure. Whether the other procedures mentioned above might under certain

circumstances or specific combinations also prove beneficial, is a question left for a further, more detailed study. In the following, we will overview the silhouette maxima obtained using variable mass scaling, as presented in Fig. 3, and the information it reveals on the general structures within our set of data.

### 3.3 Overview of the clustering results

Utilising the optimised parameters and pre-processing methods, set based on the test results, we are yet left with a number of

25 plausible solutions of mathematically almost equal quality. These solutions, shown as the bright silhouette maxima in Fig. 3, are connected to various structures in our set of pollution data. In the following we will try and interpret these data structures both from mathematical and physicochemical viewpoints.

Beginning with an overview of the favourable solutions of highest mathematical quality, we located and tabulated the maximum silhouette values obtainable for each dissimilarity metric and each number of clusters $k$. Examining the

30 corresponding silhouette value distributions (supplementary material, Fig. S.10), we set 0.45 as the prerequisite value for a solution to be included in this comparison. This translated to 12 solutions (i.e. maxima for the solution regions with silhouette > 0.45) being selected for a detailed, manual examination and interpretation. The top solutions' silhouettes and $k$ are presented in Table 2.



A brief overview of the k values associated with highest silhouette solutions *k* (Fig. 3) suggest our set of objects would best be divided either in

    a) two distinct classes, emerging from the original data without any mass scaling, or,

    b) a more complex classification leading to 6 to 10 separable classes when optimised mass scaling is applied.

We hypothesise these alternative classifications correspond to different types of structures present in the data – a two cluster structure would imply separation based on a dominant, binary-type variable, or a two-part division along a single axis (i.e. dimension, property), whereas six to ten clusters likely imply divisions along more than one dimension. In the following we first investigate and interpret the binary (two cluster) structure (Sect. 3.3.1), and subsequently aim to explain the finer, multi-dimensional structures and classifications reflected by the six to ten cluster solutions (Sect. 3.3.2).

### 3.3.1 The two cluster solution — separation by oxidation state

The two cluster ($s_m = 0$) solutions, obtained with both "cosine" and "correlation" metrics, produce the exact same bi-cluster division of objects. To understand the reasoning of this separation, we need to examine the aerosol chemical differences between the two classes implied by the division. Constructing the mass spectra from weighted cluster centres (Fig. 4) we interpret the main chemical difference between the groups is the age (i.e. oxidation level) of the aerosol. Approximated oxygen-to-carbon, "O:C", ratios can be calculated using the Aiken "ambient" parameterisation (Aiken et al., 2008) of

$$O{:}C\,(f44) = 3.82 * f44 + 0.0794, \tag{14}$$

where *f44* is the fraction of total signal observed at 44 Th. This would yield for cluster A an O:C of 0.51, branding it intermediately aged and semi-volatile (Canagaratna et al., 2015; Ng et al., 2010b), whereas cluster B's O:C of 0.16 would imply it consists of fresh, hydrocarbon dominated aerosol pollution cases situate it oxidation-wise somewhere between HOA and SV-OOA (Aiken et al., 2008; Jimenez et al., 2009; Ng et al., 2010b).

It should be noted that without mass scaling this separation is thus the most natural one (with silhouette maximum at *k* = 2; Fig. 2). The result is rather unsurprising considering the several low *m/z* (< 45 Th) oxidation-related signals (16 to 18, 29, 44 Th) usually dominating the signal fractions' distributions. However, as the result holds also for when down-weighting *m/z* 44 Th derived signals, as mentioned in Sect. 3.2.3, we believe the two-factor solution is an actual, true structure in the data, as opposed to an artefact from the AMS fragmentation table calculations. We therefore conclude the two-cluster structure represents aerosol classification into very fresh (cluster B) and relatively more aged (cluster A) groups.

### 3.3.2 Interpreting the underlying structures of higher order (k = 6…10) solutions

As the number of plausible solutions with similar magnitude (0.45 to 0.49) silhouettes for the pre-processed set of data is larger than just a few, we will not thoroughly describe all the solutions here or at this point claim one is superior to the others, but instead try to formulate a synthesis of the results, and identify the common features exhibited by the solutions.





Upon looking at all highest total silhouette value the solutions, excluding the two cluster solutions covered above and inspecting the mass spectra derived from the cluster centroids, we can find several analogous characteristics shared by essentially all the solutions. Presenting an overview of the $k = 6...10$ solutions in a tabular format (Table 2), we can begin to understand the underlying structures in common: firstly, there seem to be two ever-present, clear-cut clusters (silhouettes >

0.5) with high within-cluster silhouettes of $0.55...0.66$ and $0.47...0.57$, denoted here "strong" clusters (S-I, S-II), with minor variation in cluster population size or the resulting centroid spectra.

Secondly, there seems to always be a group of two to three "outlier" clusters (O-I to O-III), each with very unique individual mass spectra. Here we brand them outlier groups due to their small cluster populations ($n = 1...6$) and the striking dissimilarity to other observed groups (additionally quantified in Table S.1). Examining the changes in within-cluster

silhouettes, we find the inclusion of the third, singleton outlier (O-III) as its own class is a marked improvement to the solution in terms of cluster cohesion – a change also reflected in enhanced total solution quality when O-III is included.

The remaining clusters are much less pronounced (within-silhouettes typically < 0.4), and much less stable as $k$ is increased – they clearly present the most challenge for this type of an analysis. For the purposes of easy reference we term them the "weak" clusters (W-I to W-III)

Observing the weak clusters' composition, they seem to form a structure independent of that of the clear-cut and outlier groups; the sum of population over the weak clusters is rather invariable ($n = 32...35$), and only in very few cases is there disagreement among the solutions in assigning an object into strong versus weak clusters. This potentially suggests the weak structure forms a "supercluster" distinct from both the strong and the outlier clusters. The inner cohesion of this supercluster, however, seems low, as evidenced by the low within-cluster silhouettes and the interchangeability in assignments into sub-

clusters between equally good total solutions.

To examine the effect of outliers in data, we additionally tested excluding the outlier and/or the strong clustered objects and re-running the analysis for the remaining data, but the results were found to revert to an analogous situation with the same problem of silhouette-wise ambiguity and low inner cohesion of the weak sub-clusters.

From examining the within-cluster silhouette values of the clusters we would be inclined to look primarily to the Table 2

solution at $k = 8$ for correlation metric solution, due to highest mathematical solution quality (silhouette 0.49) and reasonable (silhouette > 0.25) cohesion for all of the weak clusters. However, at this point we feel to have reached the limit of what we can conclude based on the silhouette values alone, and have to also consider the aerosol chemical interpretability of the solutions.

### 3.4 Aerosol chemical interpretation of clusters

As ever when applying inherently mathematical algorithms such as PMF2/ME-2 and k-means++ to a physical or chemical experimental data, it is important to remember the algorithms are in the end only analytical tools that in the best case help in answering a particular question or gaining further understanding of the data. Their ultimate usefulness is therefore measured by the interpretability and applicability of the answer in the physical or chemical context, as much as its methodological



robustness and statistical (un)certainty. In this work the final test of our methodology is to see if we can understand the resulting cluster assignations in the context of aerosol chemistry and to interpret the clusters as air pollution types.

When interpreting aerosol mass spectra measured from ambient air, it should be kept in mind the aerosol is not only the product of the primary emission or nucleation, but also the physicochemical processes taking place post-emission. These
include notably condensation and evaporation of trace gases, as well as interaction with other aerosol types. Particularly it has been suggested the interactions between on the other hand primary and secondary aerosols, and likewise anthropogenic and natural ones and their precursors, may play a considerable role in forming and transforming the atmospheric aerosols we observe (e.g. Weber et al., 2007; Carlton et al., 2010). These interactions are poorly understood and usually not taken into account when analysing ambient observations. It seems likely, though, that these effects would hinder attempts of
classification by smearing out differences between aerosols from various different sources.

For AMS data we are fortunate to have access to years of research by the worldwide AMS users' community and the numerous studies reporting compositions of various aerosol types, which significantly helps us in understanding the cluster centroid spectra. An especially helpful information repository relevant for any AMS related mass spectral identification and comparison exercises such as the problem at hand, is the AMS spectral database (described by Ulbrich et al., 2009),
containing a total of 248 unit resolution AMS spectra from both ambient air, chamber and combustion experiments. The spectra contain examples of source-attributed aerosol types, obtained in a laboratory experiments or ambient aerosol feature extraction, various averaged mass spectra of ambient aerosols over longer periods and laboratory standards measured using the AMS. As the spectra are obtained using many different AMS variants and under very different conditions, not all the spectra contain the same variables ($m/z$) or are normalised in a standard way, which may cause uncertainty when comparing
spectra.

To help interpret our obtained clusters we calculated the similarities between the AMS spectral database specimens and the mass spectra derived from our cluster centroids. Where needed we would then refer to the specific publications describing the details of the comparison spectra of interest. As a measure of similarity we use the mass scaled spectral correlation coefficient, i.e. Pearson product-moment correlation, (Eq. 8) between the mass spectra, scaled dynamically by $(m/z)^{1.36}$. We
believe to have shown mass scaling is advantageous also for measuring the similarity of AMS spectra, as it is well known to improve spectral similarity comparisons in mass spectrometric applications (e.g. Horai et al., 2010; Kim et al., 2012; Stein and Scott, 1994), and will thus use $r_s$ and $r_s^2$.as measures of similarity between a pair of spectra. Only correlations with $p < 0.05$ are considered.

As mentioned in Sect.2.2, information on pollution event hourly times and peak wind directions were logged during the
feature extraction analysis. The summary of these diagnostics sorted according to clustering results are shown in supporting material (Sect. S.9; Fig. S.11, S.12). However, due to the small amount of objects in most clusters, sample sizes are too low for solid conclusions to be made from this auxiliary data.



### 3.4.1 The "strong" clusters – Biomass burning and sawmill pollution

The clusters we can identify, quantify and interpret with high confidence, are the strong clusters (S-I and S-II), clearly set apart by the k-means++ algorithm.

Looking at the correlations to database spectra we find the first cluster (S-I) to correlate highly ($r_s^2 = 0.85$) with the PMF

derived semi-volatile oxidised organic aerosol (SV-OOA) spectra reported by Ng and co-authors (2010a) as an average spectra of 15 ambient AMS datasets, and also correlate with several other SV-OOA mass spectra from the database. The cluster S-I mass spectrum also correlates highly with most laboratory generated boreal forest relevant secondary organic aerosols, e.g. those from α-terpinolene ($r_s^2 = 0.91$), α-terpinene (0.90), α-pinene (0.87), α-humulene (0.84) and myrcene (0.82) oxidation by ozone, reported by Bahreini et al. (2005). The spectrum also seems to closely match the biogenic

background aerosol mass spectrum generally observed at the station when anthropogenic sources are absent. This type of spectrum has also been reported previously for the site for example by Allan and co-workers (2006). As the strong plume-like nature of these air pollution events makes the possibility of a purely natural source for this aerosol type unlikely, we investigated the wind direction patterns during the peak concentration of the events classified in this category, along with location of potential local and regional aerosol sources (supporting information; Fig. S.11; Fig. S.13 to S.16). Based on this

auxiliary information we conclude the aerosol plumes likely originate from the nearby sawmills at Korkeakoski, situated some kilometres from the station, matching also the monoterpene plume observations of Liao and others (2011). Despite the chemical similarity to natural semi-volatile background aerosol in boreal forest (e.g. OOA 2 from the work of Corrigan and co-workers (2013) the PMF model does manage to reliably discriminate the sawmill plume factor from the background, so it seems evident the two mass spectra have differences. We hence label this aerosol type the "sawmill secondary organic

aerosol" (sawmill-SOA), and hypothesise it is formed via gas-to-particle conversion from the BVOC's emitted in large quantities as wood is cut and subsequently dried at the sawmills. Notable in this spectrum type is the almost complete lack of signal at 57 Th (corresponding to $C_4H_9^+$ and $C_3H_5O^+$; Mohr et al., 2012), a very typical peak to occur in most other anthropogenic AMS spectra.

The second cluster that can be easily identified is S-II. The absolutely highest correlation ($r_s^2 = 0.97$) within ambient spectra

is the PMF derived, aged, low volatile biomass burning organic aerosol (OO2-BBOA) quantified by Crippa et al. (2013) for metropolitan Paris aerosol and with other similar, highly oxidised specimen, e.g. the low volatile oxidised organic aerosol (LV-OOA; $r_s^2 = 0.93$) observed by Lanz and others (2007a) in wintertime Zurich, and suggested in their analysis to have originated from wood burning. Of the laboratory spectra it closely matches the spectra collected during a burning experiment for oak smouldering (Weimer et al., 2008; $r_s^2 = 0.85$) and burning a type of undergrowth vegetation (sage rabbit bush; $r_s^2 =$

0.88; Fire Lab at Missoula Experiment "FLAME-1" – spectra submitted to AMS spectral database by J. Kroll). We will call the S-II cluster anthropogenic low volatile oxidised organic aerosol (A-LV-OOA), since while it contains low amounts of biomass burning marker signals (m/z 60, $C_2H_4O_2^+$ and m/z 73 Th, $C_3H_5O_2^+$ fragments from the anhydrosugar levoglucosan; e.g. Elsasser et al., 2012; Cubison et al., 2011; Schneider et al., 2006), their ion concentrations remain low ($f60 + f73 < 0.01$)





and thus we are hesitant to say the aerosol is from biomass exclusively. As discussed in Sect. 2.2.1, despite limited population in the area, domestic wood burning is common in rural Finland for domestic heating during the cold season and recreational purposes (saunas, barbeques) during the warmer season, and it has been shown biomass burning smoke is rapidly oxidised upon release to the atmosphere (Hennigan et al., 2011; Cubison et al., 2011), producing low volatile aerosol

compounds in a matter of hours. It is also known (Ng et al., 2010b; Zhang et al., 2011) that upon reaching a high level oxidation, most aerosols start to resemble general LV-OOA, as they gradually lose their unique mass spectral features, making it plausible the pollution aerosols in cluster S-II are from different sources. However, compared to the highly oxidised, biogenic background LV-OOA the S-II mass spectrum exhibits the $m/z$ 57 and 60 Th anthropogenic markers and is missing the characteristic, large 53 Th peak generally reported in boreal forest biogenic background aerosol (*e.g.* OOA-1

reported by Corrigan et al., 2013). As also the plume like nature of the pollution episodes studied would imply anthropogenic sources over natural ones, we conclude S-II is almost certainly of anthropogenic origin.

The mass spectra of biomass burning and the sawmill aerosol groups, derived from the highest silhouette (0.49) solution ("corr", $k = 8$) are depicted in Fig. 5.

### 3.4.2 The "weak" clusters – Anthropogenic fresh and semi-volatile aerosols from traffic, biomass burning, cooking
**and industry**

Whether due to too low amount of observations, limits imposed by instrument SNR ratio or chemical similarity between the weak clusters, the mass spectral structures separating the weak groups' aerosols from each other is much less pronounced than the division between the strong and the outlier cluster characteristic spectra relative to other aerosol types. Although hard to judge based on this set of data alone, we feel the fault lies mostly with the last hypothesis, since the number of

20 observations related to the weak groups overall is quite large (around 40% of total) and the instrument SNR seems to enable the classification of other groups without ambiguity. From the general outlook of the weak clusters' spectra we observe many hints (low $m/z$ 44 Th signal, pronounced 55 and 57 Th peaks, distinct repeating spectral structure at 65…83 Th) pointing to the direction of fresh anthropogenic combustion originated aerosols.

The actual differentiation between AMS aerosol spectra from cooking, and traffic is notoriously hard for unit mass resolution

spectra, as discussed by Mohr and co-authors (2012), and is traditionally mostly based on the relative abundances of signals at $m/z$ 55 and 57 Th. Mass spectral differentiation between fresh BBOA and COA is even harder, as their characteristic unit-resolution spectra are near indistinguishable — we calculated a similarity of $r_s^2 = 0.83$ between (unit-resolution converted) COA and BBOA spectra from the data of Mohr et al. (2012). Also the nature of cooking fuel (e.g. wood, coal, natural gas) and use of cooking oil likely affects the resulting COA spectrum and its similarity towards either HOA or BBOA.

Looking again at the highest silhouette solution ("corr", $k = 8$), the fresh aerosol types, with lowest O:C are clusters W-II (O:C = 0.15) and W-III (0.15). Cluster W-II translates to a characteristic spectra that best correlates with hydrocarbon-like organic aerosol (HOA) reported by Ulbrich and co-workers (2009; for Pittsburg), Crippa et al. (2013; Paris), Lanz and others (2007a; Zurich), and the average HOA of 15 datasets described by Ng and co-authors. (2010b) with respective $r_s^2$'s of 0.92,



0.91, 0.90 and 0.92. Similarities with laboratory data are observed with aerosol specimen such as lubricating oil aerosol ($r_s^2$ = 0.87), diesel bus exhaust (0.90) and fuel (0.77), reported by Canagaratna et al. (2004), but notably high similarity also exist with mass spectra from burning plastic (0.96) and the various cooking experiments' aerosol products ($r_s^2$ = 0.84…0.92), described by Mohr and others (2009), as well as laboratory spectra of decanal (0.86) and hexadecanol (0.84) measured by

Alfarra et al. (2004). However, the similarities of W-II spectra to reputable cooking organic aerosol spectra, extracted from comparable ambient observations (*e.g.* Mohr et al., 2012; Crippa et al., 2013); are notably lower (0.62; 0.71), compared to the aforementioned indications this aerosol class would be related to traffic related HOA. The ratio of *m/z* 55 : *m/z* 57 signals for this aerosol type is 1.17, agreeing with findings by Mohr and co-workers (2012) for HOA.

Also the wind direction analysis combined with potential source survey (available in supporting information Sect. S.9; Fig.

S.11, S.13 to S.16), additionally points to the conclusion the source of this aerosol is in the sector with a nearby public road. We thus term W-II as hydrocarbon-like organic aerosol (HOA), in accordance with AMS aerosol naming conventions.

The other "fresh" aerosol type, W-III (Fig. 3.6) exhibits highest similarities ($r_s^2$ = 0.88, 0.86) with the aforementioned ambient cooking aerosols, measured in Barcelona (Mohr et al., 2012) and Paris (Crippa et al., 2013) while correlating markedly less (0.53…0.72) with the HOA spectra of the database. Laboratory spectrum matches are with charbroiling (0.72;

Lanz et al., 2007a) β-caryophyllene (0.87) β-pinene (0.75; Bahreini et al., 2005), the former sesquiterpene being an important constituent in many essential plant oils used in cooking. Moderate correlations ($r_s^2$ = 0.56…0.70) are found with Mohr et al. (2012) cooking aerosol specimen and the various smoke chamber spectra from FLAME-1 (0.36…0.79) mass spectra (Fire Lab at Missoula Experiment – spectra submitted to AMS spectral database by J. Kroll). Signal ratio *m/z* 55 : *m/z* 57 for the W-III spectrum is 3.14, which when interpreted in accordance with the COA estimation method introduced by Mohr and co-

authors (2012), suggests this aerosol type would be cooking-related. We therefore label the W-III cluster as cooking organic aerosol (COA). However, in the end, due to the close similarity of COA and BBOA (Mohr et al., 2012), we cannot rule out the possibility of fresh biomass burning or combustion aerosol from barbeques also contributing to this mixed class of observations.

Separating both of these fresh two sub-classes from the "weak supercluster" leaves us with the semi-volatiles species in form

of one to three clusters. The solution with one semi-volatile aerosol pollution type, W-I, in ("corr", $k$ = 8) is mathematically the most robust one.

W-I pollution type exhibits mixed mass spectral characteristics between the HOA and COA types (Fig. 6). The main difference with the former two clusters' spectra is the spectra from the remaining part of the weak clusters' (W-X) group implies considerably higher oxidation state (estimated O:C ratio of 0.40; compared to 0.18 and 0.15 for HOA and COA;

Aiken et al., 2008; Eq. (14)). The library spectra similarity examination brands this aerosol as general semi-volatile oxidised organic aerosol (SV-OOA), with closest similarity to SV-OOA observed in Barcelona ($r_s^2$ = 0.91; Mohr et al., 2012) and Pasadena ($r_s^2$ = 0.88; Hersey et al., 2011). The similarities to ambient urban aerosols, HOA and COA specimen as well as traffic, burning and cooking related laboratory spectra are generally moderate to high (typically 0.5…0.8) but with no real pointers to a single, dominant type of origin over the others. The ratio of *m/z* 55 : *m/z* 57 of 1.57 is between that of the HOA





(1.17) and COA (3.14) spectra  (Mohr et al., 2012) and the higher $m/z$ range (45 to 100 Th) seem to offer little in terms of features distinct from COA and HOA. We brand W-III as A-SV-OOA, for anthropogenic semi-volatile oxidised organic aerosol, to separate it from the biogenic and natural SV-OOA types such as semi-volatile forest background or sawmill-SOA aerosols, as the close connection to combustion-related aerosol types seems evident based on the (dis)similarities between

the clusters. We hypothesise this aerosol type is a mixture of anthropogenic aerosols from various origins, such as traffic, cooking, and possibly industrial processes, the common feature of which is that it has been subjected to some oxidation and mixing, smearing out the characteristic features of more distinct classes of aerosols such as the fresh HOA and COA types. However, we will also present here a further-going interpretation based on the $k = 10$ ("corr") solution with three separate A-SV-OOA factors: we suggest these three classes could be interpreted as source-specific anthropogenic SV-OOA types. We

hypothesise the differences between the fresh aerosol types, sorted according to their emission source, are not yet completely smeared out by intermediate level of oxidation. This would allow k-means++ to differentiate (albeit with much less confidence) between the A-SV-OOA types, resulting in differentiation based on origin either from traffic SV(HOA), cooking SV(COA) or biomass burning SV(BBOA), shown in Fig. 7.

This interpretation is indeed supported to some extent by the correlation examination against HOA, COA and BBOA spectra

of the AMS spectral database (Supplementary information; Table S.3), and in case of SV(HOA), the only group with a moderate number of observations ($n = 11$) also by the wind direction analysis, pointing to the south-to-west sector with the main nearby roads as the sector of origin (supporting information; Fig.s S-11 and S-13). To corroborate this finer source specific differentiation of A-SV-OOA, however, a larger amount of observations would certainly be beneficial.

### 3.4.3 The outliers – amine compounds from biogenic sources?

While the spectra examined thus far seem interpretable in the "traditional" framework of AMS aerosol types classification (LV-OOA, SV-OOA, BBOA, HOA, COA), the outlier clusters do not fit these conventional categories. There are also no spectra matching our observations in the AMS spectral library. We therefore additionally examine the spectral features and compare them to observations in other mass spectrometry literature.

To begin with, we note the distinctive feature of all the outlier clusters' mass spectra are rather "exotic", at least in AMS

context, with peaks at 58, (72), 86 and 100 Th (Fig. 8). These even molecular masses are relatively rare to be observed in the AMS organic spectra due to the nitrogen rule implying the presence of a nitrogen atom. The homologous ion series of amine compounds ($C_nH_{2n+2}N^+$) yields masses 30, 44, 58, 72, 86, 100 Th (Kraj et al., 2008), exactly matching the peaks not obscured by other organic ions, which suggests presence of various amine compounds. We also came across a laboratory study by Rollins and co-workers (2010), which reported increased signals for $m/z$ 58 and 86 Th also for when synthesised

hydroxynitrates were measured with an AMS. However, their spectra seem to be dominated by peaks not observed in our outlier spectra, and the said nitrate series ($C_nH_{2n+2}N^+$) only forms a minor part of the whole signal reported by Rollins and others. Also Wolf et al. (2015) report some atmospheric bacteria related to ice-nucleation producing a 70 eV EI MS signal at 86 Th ($C_5H_{12}N^+$), but this sample lacks the other signals present in our series.





We additionally note that similar homologous series exist for some aliphatic ketones ($CR_2=C(OH)R^+$) following McLafferty rearrangement, producing a unit mass series at 58, 72, 86 and 100 Th (McLafferty, 1959). Conclusively differentiating between these two organic groups would benefit from high-resolution ToF data, which we unfortunately do not have available at this time. While we cannot definitely rule out the possibility of high concentrations of ketone compounds, we do

not find references to this type of observations in the aerosol mass spectrometry related literature.

Conversely, there are overwhelming numbers of observations of amines in in aerosol phase — Ge and co-authors (2011) calculated in their review article a grand total of 67 aerosol phase amine observations (1972 to 2009). Within AMS measurements, amines have been postulated from unit resolution mass spectra (Aiken et al., 2009) and confirmed via high

mass resolution analysis (Huffman et al., 2009; Sun et al., 2011) in studies conducted in the heavily populated and industrialised Megacities of New York, and Mexico City. Also Allan and co-workers (2006), performing the first quadrupole AMS measurements at the SMEAR II site in 2003 already speculated on the possibility of amines explaining the "extra" nitrate signal at 30 Th, not explicable by ammonium nitrate alone. The amines' contribution at 30 Th peak corresponds to the $NH_2CH_2^+$ ion, but is often obscured in unit-resolution data by other organic fragments and the $NO^+$ fragment from common

$NH_4NO_3$.

Of the aforementioned studies only the amine containing aerosol from New York was available, submitted to the AMS high resolution database (Ulbrich et al., 2009; http://cires1.colorado.edu/jimenez-group/HRAMSsd/) with reference to after a later similar analysis by Docherty et al. (2011). This high-resolution AMS spectra was translated to unit mass resolution and compared with our samples. A moderate similarity ($r_s^2 = 0.67$) was found between the library specimen and our cluster O-III,

lending some confidence to the assertion of similarity. Although the other amine spectra were unavailable for mathematical correlation check, we note the general spectra of the aerosols reported by Huffman and others. (2009) for Mexico City and Sun et al. (2011) for New York exhibit some similar features to the spectra of O-I and O-III clusters, namely increased $m/z$ 58, 72 and 86 Th signals, but in different relative fractions.

The aerosol described by Sun et al. has the major nitrate containing peaks at $m/z$ (56), 58, 59 and 72 Th, but shows only a

small peak at 86 Th and no significant contribution at 100 Th. The two Mexico City spectra reported by Huffman et al. (2009) include major peaks at 58 and 86 Th, but little to no contribution at 100 Th. By visual inspection the Aiken et al. (2009) amine aerosol specimen doesn't seem to contain any of the peaks discussed here, so we consider it not to be a relevant reference in this particular case.

Additionally, some very similar spectra we encountered in literature were from those laboratory SOA formation study by

Murphy et al. (2007), who measured using an AMS, secondary aerosol generated from various aliphatic amines. The spectra they report for trimethylamine photo-oxidation product aerosol has multiple similar features at 58, 86 and 100 Th, albeit in different signal fractions ($m/z$ 86 Th signal > $m/z$ 100 Th signal) to ours. Looking up the 70 eV electron impact ionisation spectrum of trimethylamine, available from the U.S. National Institute of Standards and Technology (NIST), we find a specimen with the same peaks as our but again with different ratios for $m/z$ 58, 72, 86 and 100 Th.





Closest match for the O-I and O-II clusters within the AMS spectral database was the marine aerosol ($r_s^2 = 0.50$ O-I; 0.68 O-II) reported by Chang and co-authors (2011) for Arctic Ocean marine biogenic aerosol. It does contain a small peak at 58 Th, but only low signals at 86 or 100 Th.

While the final decision on the sources and origins of the outlier clusters' spectra remains controversial, we believe the likely explanation for the outlier spectra are the amine compounds, a hypothesis based on the confirmed AMS amine observations in the aerosol phase, and the laboratory tests of Murphy and co-workers (2007), along with the lack of credible alternative accounts or explanations for understanding the *m/z* peaks observed. Therefore we name the outlier I-III peaks "amine-58" (O-I), "amine-100" (O-II) and "amine-86" (O-III) respective to their major characteristic peaks, most likely corresponding to fragment ions with elemental composition $CH_4N^+$ (at 30 Th), $C_2H_6N^+$ (44 Th), $C_3H_8N^+$ (58 Th), $C_4H_{10}N^+$ (72 Th), $C_5H_{12}N^+$ (86 Th) and $C_6H_{14}N^+$ (at 100 Th).

The aerosol phase amine sources have thus far mostly been attributed to either local industrial pollution or marine biological production (see Ge et al., 2011 for a review of observations). In our case both of these sources would be surprising considering the inland location and the scarcity of nearby industrial plants, along with the apparent seasonal dependence of the observations (only observed in the springtime measurements). However we cannot rule them out at this point. As additional hypotheses for the origin we offer the following; it would be conceivable the observations correspond to SOA formation from volatilised and possibly oxidised amine compounds, or amine salts, with the potential sources being

1. Amines from biodegradation of organic material such as leaves and needles, released from being trapped below the layer of snow and evaporated from the solar-heated ground surface at and after snowmelt. Amines are known to be common products of biomaterial degradation processes and to be volatile (Kieloaho et al., 2013; Kuhn et al., 2011). Alkylamines are also known to be produced at the forest floor and their concentrations are found to be temperature dependent and peaking during autumn forest litterfall (Kieloaho et al., 2013). Our AMS measurements coincided with the snow melt period at the site, but to our knowledge no data amine data from snowmelt period at SMEAR II exists, regrettably.

2. Manure application in crop fields has been shown to be a potential source of amines (Schade and Crutzen, 1995; Ge et al., 2011). It would be plausible some nearby field that has been fertilised with manure, would release a considerable amount of volatile amine compounds upon drying. This hypothesis has been questioned, however, by the findings of Kuhn et al. (2011) who instead conclude the amine emissions are more likely from the animals' exhalation and feeding. There exists a cattle farm some two kilometres to the east of the field station and some agricultural fields closer by (supporting information Fig. S.14).

3. A nearby patch of forest of 0.8 hectares of area, at a distance of 300 to 500 meters from the site, was cut clear a month prior to the start of our measurements (Virkkula et al., 2014). Amines have been found to be emitted from tree trunks and needles of live coniferous trees of the boreal zone (Kieloaho et al., 2013), and it would therefore be imaginable the process of clear-cutting a forest stand using a harvester vehicle and the subsequent transportation of



the trunks away from the site would cause considerable emissions of volatile organic compounds, including monoterpenes and amines for several weeks after the process.

However, to confirm the amines' presence, identify the specific compounds and finally decide between these and the earlier hypothesis of the proposed origins, more comprehensive experimental measurements and analysis are likely required.

## 3.5 Interpretation of spectral structures and main dimensions defining the pollution types

Below we try to summarise what we consider the most important dimensions or axes, on which the more complex ($k = 6…10$) classifications would be based, and their interpretation in an aerosol chemical framework.

### 3.5.1 Oxidation level and aerosol age

Traditional AMS spectral analysis revolves around studying the process of oxidation, or aging of an aerosol particle in the atmosphere. The oxidation process depends on particle chemical structure, number and type of oxidant radicals available and the time spent in the atmosphere, so it is highly variable and difficult to model. From this branch of study and the connection of volatility to oxidation level (Donahue et al., 2011; Donahue et al., 2012; Kroll et al., 2011; Jimenez et al., 2009) originate also some of the "standard" labels for atmospheric processed aerosol types (LV-OOA, SV-OOA). It has been known for a long time in the AMS community, that mass spectra peaks such as $m/z$ 43 Th ($C_3H_7^+$ fragment from alkyl group molecules and $C_2H_3O^+$ from non-acid organic oxidation products; e.g. Ng et al., 2011a) and 44 Th ($CO_2^+$; common fragment from carboxylic acids; Duplissy et al., 2011), as well as their relative contributions, are good indicators for oxidation (Aiken et al., 2007; Ng et al., 2011a; Canagaratna et al., 2015). Upon aging the fraction of organic aerosol signal observed at 44 Th ($f44$) and O:C ratio of a particle increase, and the marker for fresh emissions, $m/z$ 43 Th signal goes down along with most high mass (> 45 Th) signals. Agreeing with the clear separation of aerosol types by age found in the clustering solutions of this work, the "oxidation axis" is clearly one of the main dimensions along which cluster borders are drawn.

### 3.5.2 Aerosol source specific characteristics

The other axes for cluster separation seem to relate to their source-specific fingerprints. The results presented in Sect. 3.1 and 3.2, and particularly the solution diagnostics values shown in Table 2, suggest that there are one or more source related divisions resulting in a fairly clear-cut separation of clusters. One such clear division seems to be between the anthropogenic aerosol groups considered primary, and thus usually originated from a combustion process (such as biomass or fossil fuel burning, combustion engines exhaust or aerosol formed in high-temperature cooking), and the secondary aerosol from particle conversion biogenic organic vapours (albeit in our case from "anthropogenic sources" in form of the sawmills). In our case this distinction separates in a clear-cut manner the sawmill secondary organic aerosol (cluster S-I) from other aerosols of similar age and oxidation from different sources (especially W-I). A short examination on a potential S-I spectral marker at $m/z$ 53 Th can be found in the supplementary information (Sect. S.11, Fig. S.17)





The structure the most difficult to explain conclusively is the set mass spectral features setting apart the various components of the weak cluster structure observed. The separation of fresh HOA from COA and BBOA has been discussed and characterised in many studies (*e.g.* Crippa et al., 2013; Mohr et al., 2009), but in practise classifying these aerosols in an unambiguous manner remains troublesome. It does, however, seem clear from the results presented here, that the *f55:f57*

ratio is indeed a viable indicator of a dimension separating HOA pollution type (low *f55:f57*) from COA and BBOA, as suggested by e.g. Mohr and Crippa and co-authors. We note the *f55:f57* values derived from the clustering solution, 1.17 for HOA and 3.14 for COA, match well with the estimates given by Mohr (0.9 ± 0.2 for HOA; 3.0 ± 0.7 for COA). Furthermore, there also appear to be additional, equally definitive indicators available in the higher masses, as discussed in the supplementary (S.11; Fig. S.18)

As the important separation of the sawmill-SOA cluster (S-I) also happens to be clearly reflected in the *f55:f57* dimension, due to the very low *f57* signal in its centroid mass spectrum, we adopt this axis selection along with the oxidation axis (reflected by estimated O:C)  as a basis for representing the clustering solution in a simplified way. This results in a two-dimensional projection of the 125-dimensional data structure (Fig. 9). It should be underlined, that this representation is a crude simplification of the actual solution, aimed at providing at least some visualisation of the tremendously more complex

spatial structure. Consequently, many of the potentially more complex structures located higher up on the *m/z* scale equally driving the solution are not shown, which explains why some points seem to be out-of-place in the two-dimensional projection. With that said, the solution does seem to make a lot of sense, and we can see the clusters are relatively well defined.

For this set of observations we did not obtain separate a distinct (fresh) BBOA cluster, so we were unable to evaluate the

difference between BBOA and COA. As for the more controversial classification of A-SV-OOA subtypes, the separation can be visualised in the (*f55:f57*, *f60+f73*) space (Fig. 10), *f60* and *f73* corresponding to the expected biomass burning "axis" (Cubison et al., 2011; Elsasser et al., 2012; Schneider et al., 2006). The low cohesion of especially A-SV(COA) and A-SV(BBOA) clusters is likely due to both a) very few observations available and b) scarcity of clear mass spectral differences between the groups.

**3.5.3 "Exotic" variables specific to outlier observations and groups**

In addition to these more traditional fingerprints in the AMS spectra, in this case we also have outlier observations, distinguishable by their unusual high mass (*m/z* 58, [76], 86, 100 Th) signals. It seems evident the dimension separating these groups would correspond to these specific variables. This reasoning is also supported by visualisation of the outlier spectra (Fig. 11) in an appropriate 2-d space (e.g. *f86+f100* vs *f58*).

**3.6 Estimating the natural variability within the aerosol types**

Finally, we examine briefly the intra-cluster variabilities, translating to inferred mass spectral variability within the aerosol types. While we feel it would be dangerous to claim the variation within the spectra of a specific group can be directly





understood as the natural variability of that aerosol type at this site, we propose it can be considered as an upper limit estimate of this variability, since the within-cluster variation is caused both by the actual variability in the natural aerosol, and the uncertainty induced by its measurement and analysis. Overall, the effects of instrument (white) noise is filtered in the feature extraction (PMF) phase, and the effects of possible misclassifications in clustering are likely limited to borderline

(between cluster) cases, that have minimal influence on the final spectra due to the silhouette-based posteriori weighting. The rotational ambiguity of PMF remains an issue, and while we have done our best to find the cleanest possible separation of the pollution and background spectra, some degree of uncertainty is unavoidable. Although there some tools have been proposed to assess the rotational sensitivity (e.g. bootstrapping; Norris et al., 2008; Tibshirani et al., 2001), the exact level of mass spectral uncertainty arising from the rotational ambiguity remains difficult to quantify. Also as the standard k-means

does not utilise information of uncertainties of input objects, a profound error analysis would require more advanced classification tools. We note the uncertainty estimates of PMF results is a topic still requiring attention, as highlighted by Reff and co-authors (2007), and the field of AMS PMF would likely benefit from development of further easy-to-approach statistical tools. Nevertheless, considering there exist very few if any statistically well founded estimates for this type of aerosol variability, we propose that in absence of more reliable results, the variabilities implied by this study can be used as

an indicator of what the likely magnitude of the underlying natural variability within the observed classes of aerosols at a site like this.

We examined the within-cluster variabilities of the aerosol types studied, and calculated silhouette weighted standard deviations as a function of $m/z$ ratio, which was then fitted with constant, linear and exponential (quadratic) regression models. An example of such a parameterisation is shown in Fig. 12, and the model parameters for all clusters (in the "corr" k

= 8 solution) are given in Table 3.

.

The variability parameters are especially important for (partially or fully) constrained factor analysis, such as techniques utilising the ME-2 algorithm. In *e.g.* the most commonly approach used in the Source Finder (SoFi), a single value ("a-value"; Canonaco et al., 2013), it is typical to restrict allowed spectral variation to a certain fraction of the reference spectra,

applied uniformly across all $m/z$ ratios. Based on this work we find the "a-value approach" may not be the optimal way to restrict spectral variation allowed in factorisation models such as the ME-2 driven constrained PMF, and that $m/z$ dependent parametrisations would better represent the actual natural variability that should be accommodated by the model. Ultimately, pulling approaches (Canonaco et al., 2013; Paatero and Hopke, 2009) might prove preferable to hard limit constraints for variation. Nevertheless, if still opting for the use of a constant a-value, our results imply the natural variability within an

aerosol type may be significantly larger than what is often allowed in conjunction with the constrained PMF/ME-2 (e.g. Crippa et al., 2014).





## 4 Conclusions

While advanced data analytical techniques, such as PMF, have already been widely adopted for AMS data reduction and feature extraction, the application of similar chemometric methods for AMS spectra identification and classification is as of yet an uncommon sight.

In this study we make a pitch for adopting some of the tried and tested statistical methods from other mass spectrometric fields into the analysis of AMS results. As a practical example we present a case of applying simple clustering to a set of AMS pollution spectra, and show even a simple algorithm such as the k-means++ can, with proper optimisation, match and reproduce the traditional "expert classification" of AMS aerosol types unsupervised (*i.e.* without *a priori* training).

Clustering as a method is especially sensitive to certain parameters; the algorithm used, data pre-processing (scaling) and the
dissimilarity measure (distance metric) used for the objects' spatial representation. In this work we compared the performance of some of the most basic distance "metrics" in k-means++ clustering for our example data, along with some suggested data pre-processing (scaling) methods. At least in the context of this limited case, the "[Pearson] correlation" metric seems slightly preferential as a measure of spectral (dis)similarity, followed close by [dot product] cosine and squared Euclidean dissimilarity measures – at least when the AMS mass spectra are normalised. For representing spectral similarity
of un-normalised data, we suggest either cosine or correlation metrics due to mathematical considerations (*i.e.* multiplication invariance).

Optimised mass scaling, that is weighting the mass spectra signals by an exponential function of their *m/z* ratios, seems beneficial for unsupervised classification of AMS aerosol types. Based on our example set of data we suggest scaling the signal variables at each mass-to-charge by an exponential weight $(m/z)^{s_m}$ of $s_m = 1.36 \pm 0.24$. Contrarily, intensity scaling,
or scaling the MS variables (signals) by their root function appears to be detrimental for the structure of our spectral dataset. We hypothesise this may be due to our spectra being normalised to unity and generally not being overtly dominated by any individual signals – unlike spectra in many soft ionisation MS applications – potentially upscaling general instrument noise more than the informative minor signals.

Without scaling as a pre-processing step, k-means++ produces a differentiation between oxidised and fresh(er) aerosol
samples. Up-weighting higher *m/z* signals allows for classification in the framework of source-specific AMS organic aerosol "sub-categories" such as differentiating between HOA and COA, strongly indicating much of the information needed for this classification resides among the higher up *m/z* variables. We thus suggest taking this piece information into consideration when interpreting and classifying AMS spectra, either manually or in applying a machine-learning approach. Exploring similar mass scaling in connection with comparable statistical analysis methods may prove useful especially in applications
where data weighting is already commonplace and easy to implement.

Limiting the role of PMF to solving simple cases with few factors, allowed for (almost) unambiguous identification of the physically meaningful rotation, which best temporally separates the pollution plume from the background aerosol, with minimal need for human analysts' expert judgement. Similarly, applying computer-aided, unsupervised classification any



result should be more or less free of analyst bias, when deciding the classifications of mass spectra to organic aerosol subtypes. Naturally, this does not excuse the human analyst from the final responsibility of physicochemical interpretation, comparison and evaluation of the mathematical solutions produced by any classification algorithm. It should also be noted the first phases of the task, of manually identifying pollution events and performing the feature extraction step for each of the

events individually is very labour-intensive – hence the process should ideally be made more automatic.

Despite this cost, compared to alternative approaches such as using PMF to directly extract SOA subtypes and identifying the correct rotation from the resulting solution space, we suggest the methodology presented here has several advantages: 1) due to limiting the role of the analyst mainly to deciding the correct number of aerosol classes we largely avoid the PMF's Achilles heel of rotational ambiguity and the need for major expert judgement in selection of rotation. 2) From the clustering

solution it is straightforward to derive a quantitative estimate for the uncertainty of a resulting reference spectrum – a piece of information which has direct use and value when applying the reference spectra in e.g. ME-2 analysis. Finally, 3) by analysing air pollution cases individually we can also identify and extract minor sources and identify outlier aerosol types, which fall way under "PMF's limit of detection", of explaining approximately 5% of the variability of the total aerosol mass (Ulbrich et al., 2009). These outlier groups may ultimately prove important and offer new scientific information, as

exemplified by the observation of suspected amine compounds presented in our results.

In our example of applying feature extraction (PMF) and unsupervised classification (k-means++) to a set of AMS data, we could produce reference spectra and their variability estimates for local pollution "archetypes". Aerosol chemical interpretation of the results from our testbed set of data from a background, boreal forest station (SMEAR II) suggests the main dimensions or "axes", driving the classification relate to a) oxidation state reflecting aerosol aging b) source types,

whether representing spectral structures of various combustion source types (traffic, cooking, biomass burning) or characteristics of aerosol formed from biogenics through gas-to-particle conversion  c) "exotic" variables characteristic of outlier observations and "outlier groups" (from the perspective of traditional AMS aerosols' classifications).

Ultimately, we hope to have demonstrated that statistics-based, computer-aided classification of AMS spectra seems promising, and in that the differences and characteristic features of mass spectra can indeed be parameterised for an

increasingly machine-learning oriented approach to AMS advanced data analysis.

**Data availability**

Data used is available upon request from the authors. Cluster centroid spectra will be made available in the AMS Spectral Database (http://cires1.colorado.edu/jimenez-group/AMSsd/) upon publication.

**Disclaimer**

The authors declare that they have no conflict of interest.





**Acknowledgements**

We acknowledge the funding from the following programmes: European Commission FP6 projects EUCAARI (036833-2), FP7 ACTRIS (262254), the Horizon 2020 project ACTRIS-2 (654109), ERC Grant COALA (638703), the Finnish COE project CRAICC (272041) and the Academy of Finland COE in Atmospheric Science (2008 - 2019).

We wish to thank the Hyytiälä and Helsinki technical staff (Pasi Aalto, Erkki Siivola, Frans Korhonen, Heikki Laakso, Toivo Pohja, Veijo Hiltunen and Janne Levula) for support with experimental measurements and the AMS users' community for valuable feedback on the data analytical topics.

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



**Figures & Tables**

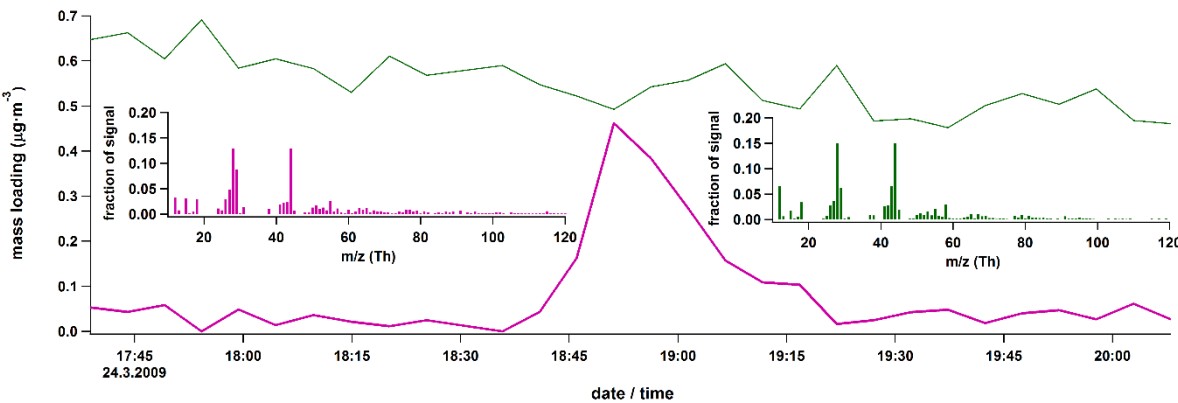

5    **Panel 1a. Pollution plume characteristic spectra (magenta) extraction. A simple, clear case with stable background (green).**

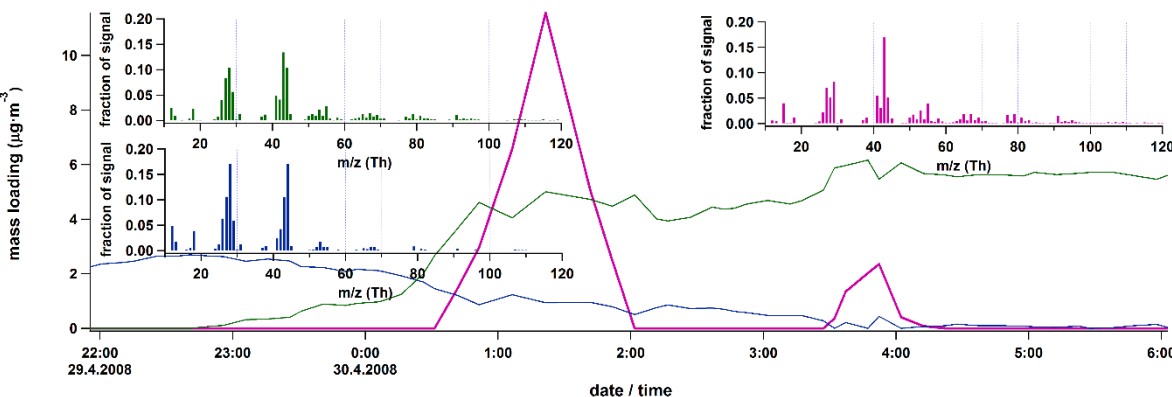

**Panel 1b. An example of a well-resolved extraction with a repeating pollution plume (magenta) and two changing background factors (in green and blue).**

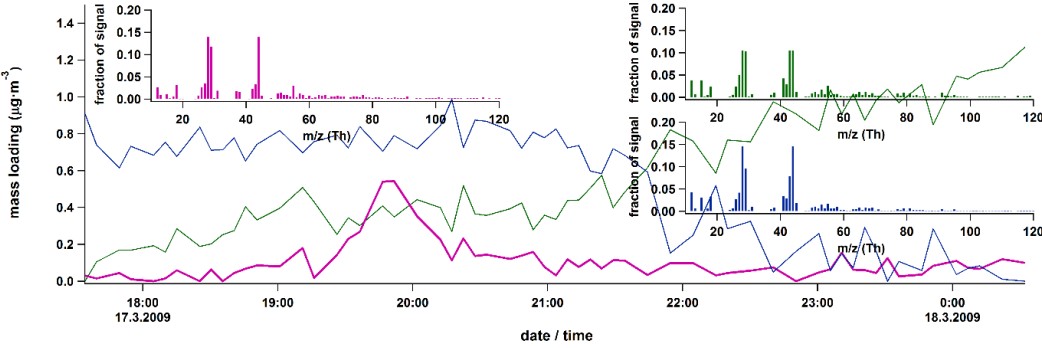

10    **Panel 1c. An extraction of a weaker pollution plume (magenta) factor from two temporally changing background factors (green and blue)**





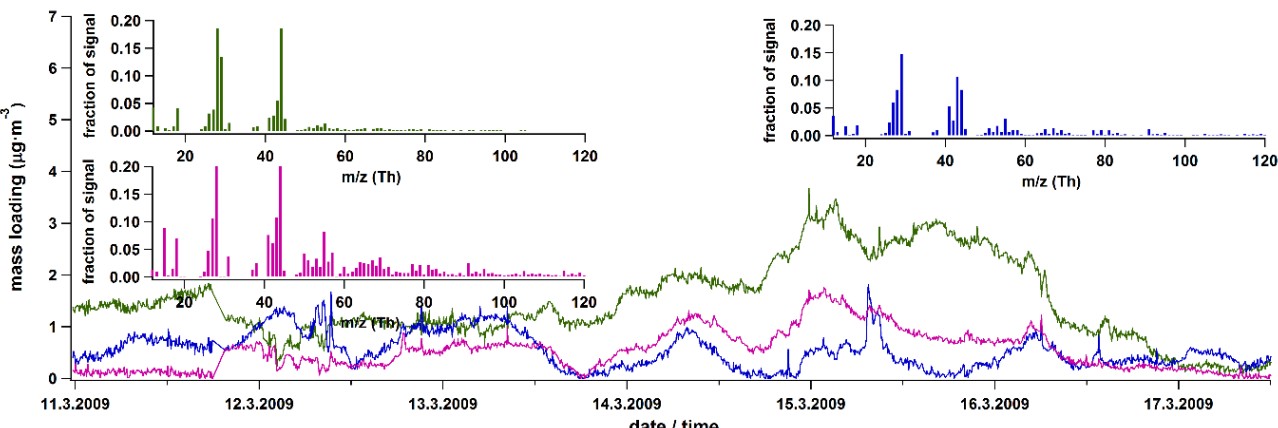

**Panel 1d. Extraction of characteristic spectra from a very complex pollution case. The two background factors are shown in green and blue. Additionally to the pollution factor shown here (magenta), three additional pollution factors (displayed in Panel 1d) were separated,.**

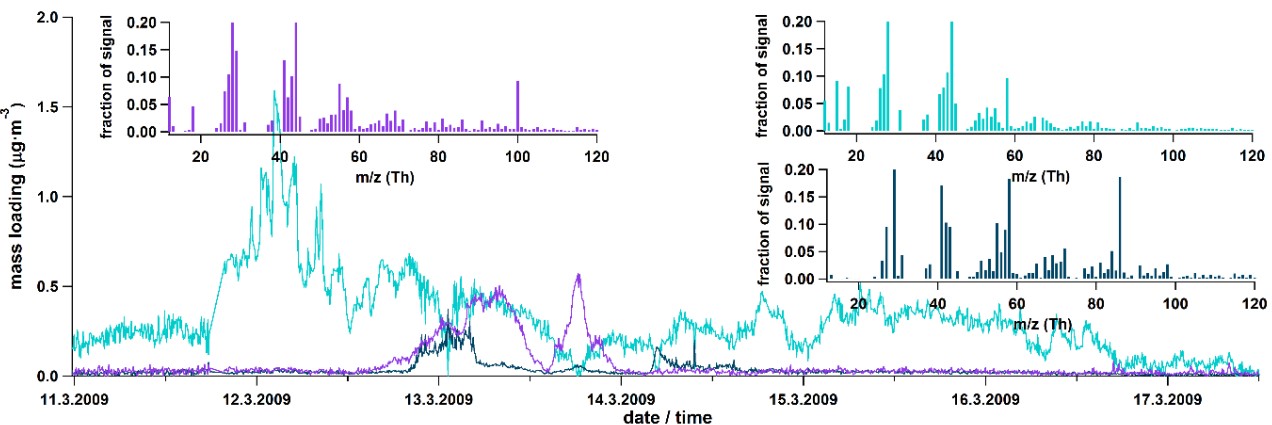

**Panel 1e. Three additional pollution factors extracted from the case described in Panel 1d.**

**Fig. 1. Examples of time series of extracted PMF factors for four different pollution cases, ranging from simple (panel a) to very complex cases (panel d & e). In panel b the factors are marginally correlated, but the separation is still considered clear. The event shown in panels d and e is a borderline accepted case due to its long duration (criterion 1; Sect. 2.2.2) and presence several pollution types (criterion 2), but was eventually accepted to the analysis due to the clear temporal and chemical separation of the pollution factors.**





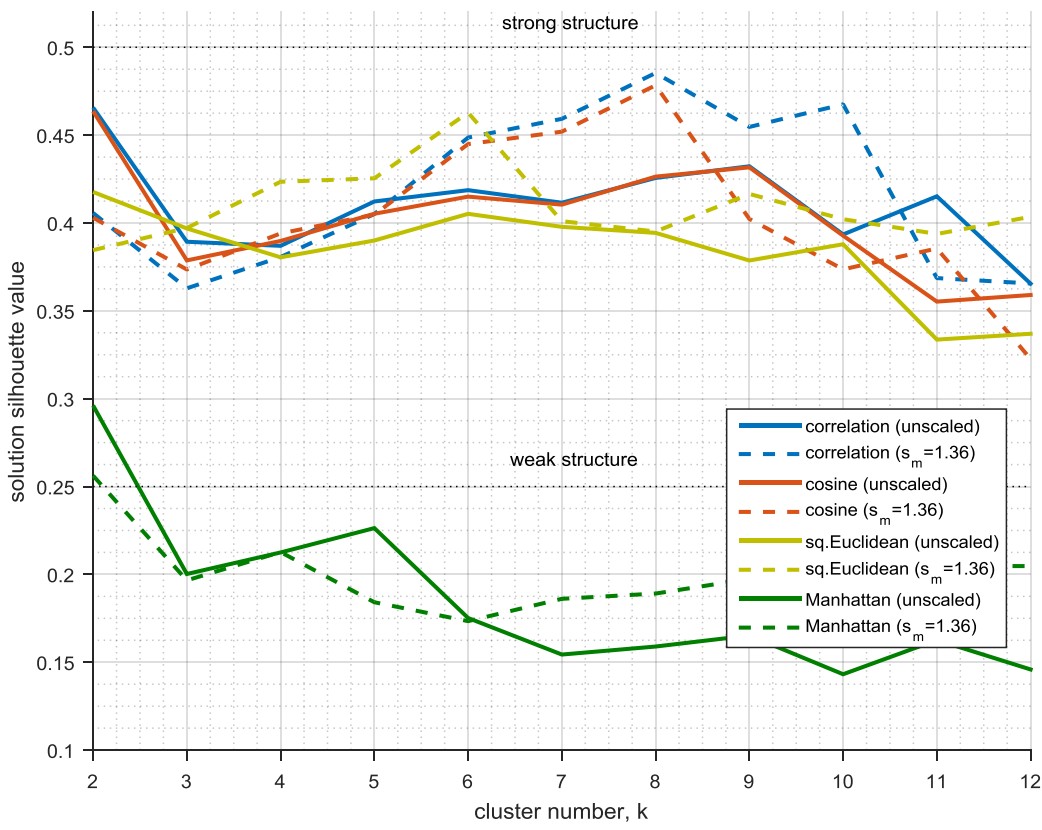

**Fig. 2.** Solution silhouette value of clustering solutions for k = 2 to k = 20, for the four dissimilarity metrics (colour coded). The solid lines depict solution silhouette values for unscaled data and the dashed lines represent the solution qualities for when data weighting is applied (non-optimised mass scaling, $s_m$ = 1.36; Sect. 2.3.3). The 0.25 and 0.50 limit (dotted black lines) indicate lower limits above which we could expect "weak" (silhouette 0.25) and strong (0.50) structures to exist in the data (Kaufman and Rousseeuw, 2009; Sect. 2.3.4).





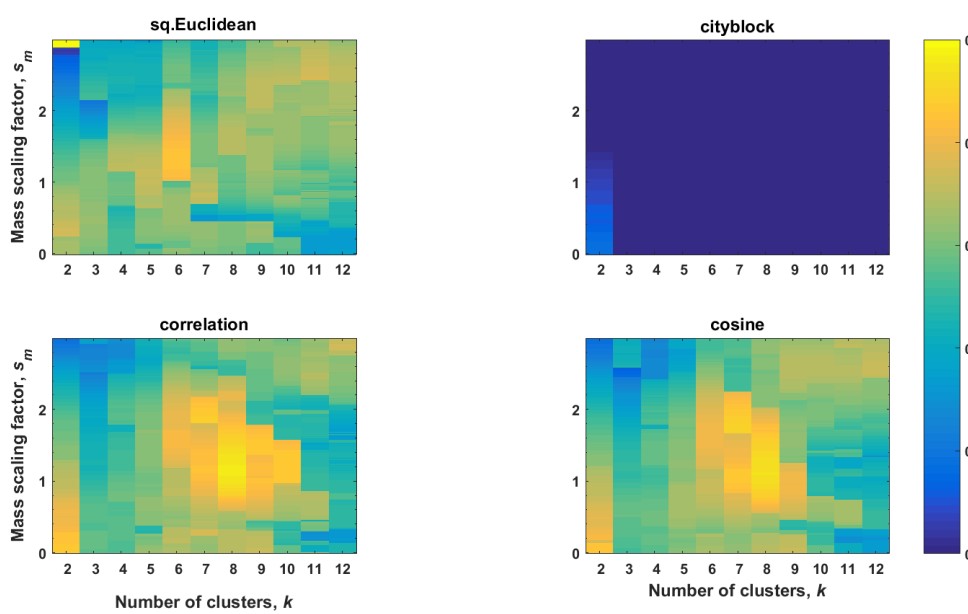

**Fig. 3. A contour plot for a field of solution silhouette values for the studied dissimilarity metrics, as a function of cluster size $k$ (x-axis) and mass scaling factor $s_m$ (y-axis). The maximum near squared Euclidean $k = 2$, $s_m \approx 3$ indicate a solution with high silhouette, but this solution only separates one outlier cluster ($n = 3$; cluster O-II, explained in the next section) from the other objects, so this solution was not considered separately.**

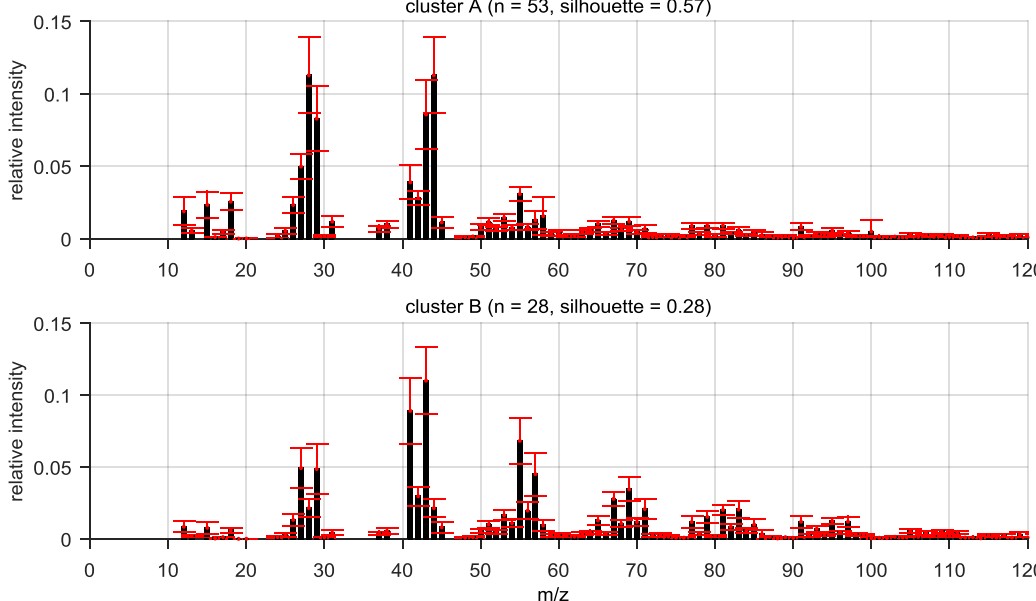

**Fig. 4. Weighted cluster centroid spectra for solution $k = 2$ ("corr" and "cos") for the non-pre-processed dataset. The $k = 2$ division appears to be driven mainly by age of the aerosol; cluster A corresponding to aged and cluster B to fresh aerosol. The error bars denote weighted within-cluster standard deviation.**





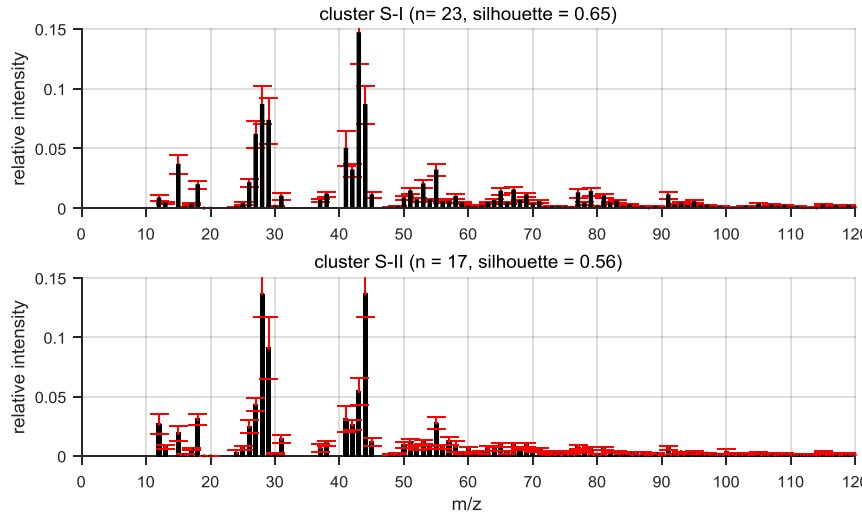

**Fig. 5. Mass spectra derived from ["corr", $k = 8$] clusters S-I (sawmill-SOA) and S-II (A-LV-OOA). Error bars indicate within-cluster variability (silhouette-weighted within-cluster standard deviation).**

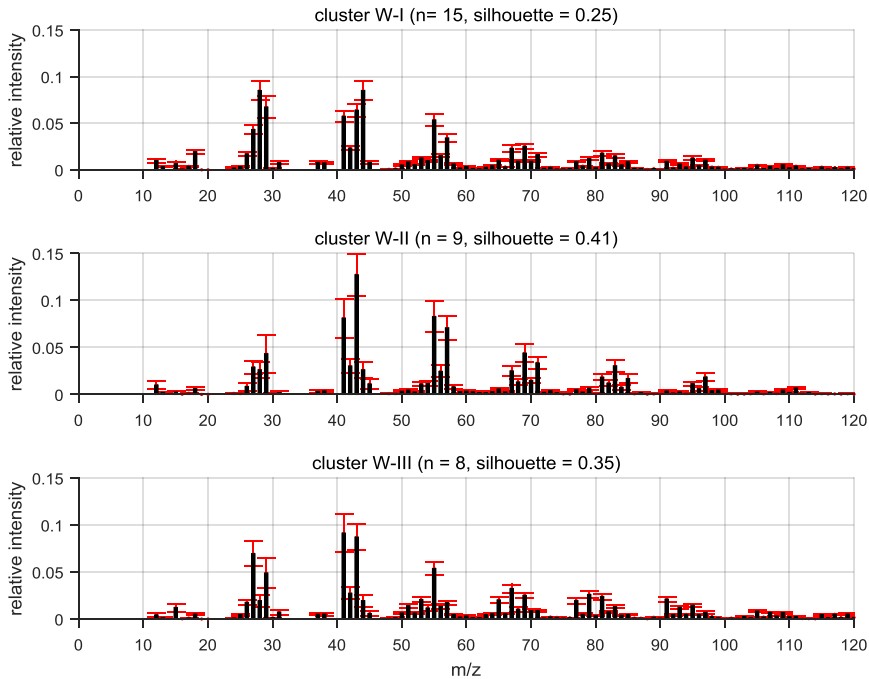

5     **Fig. 6. Mass spectra corresponding to clusters W-I (A-SV-OOA), W-II (HOA) and W-III (COA). Error bars denote within-cluster variability (silhouette-weighted within-cluster standard deviation). Solution for "corr", $k = 8$.**





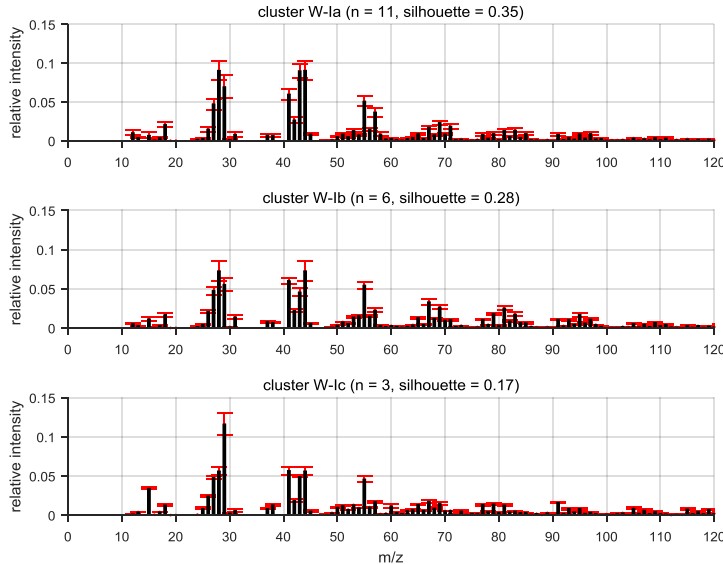

**Fig. 7. Mass spectra for the "corr" $k = 10$ solution, dividing the intermediate oxidised objects into three groups, labelled here, according to their proposed sources, as W-Ia (SV(HOA)), W-Ib (SV(COA)) and W-Ic (SV(BBOA)). Error bars represent within-cluster variability (silhouette-weighted within-cluster standard deviation).**

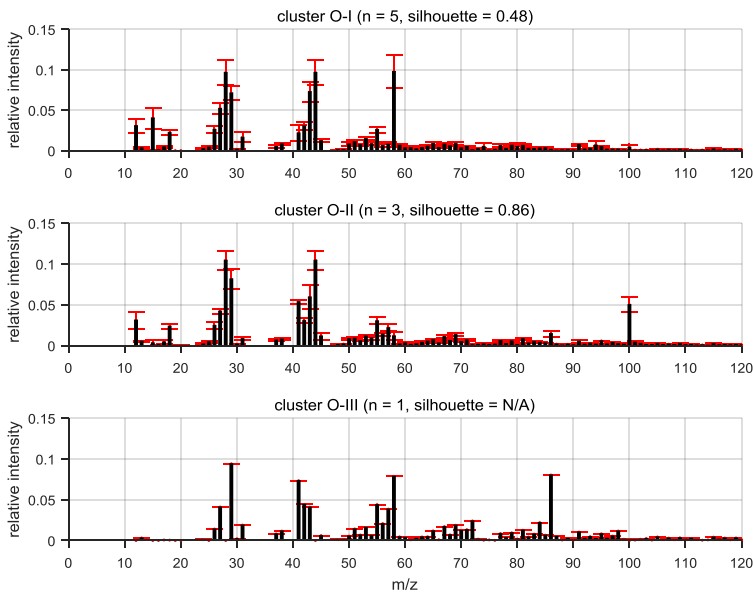

**Fig. 8. Mass spectra corresponding to weighted cluster centroids ("corr", $k = 8$) for groups O-I ("amine-58")), O-II ("amine-100") and O-III ("amine-86"). Error bars denote within-cluster variability (silhouette-weighted within-cluster standard deviation, unavailable for the singleton O-III).**





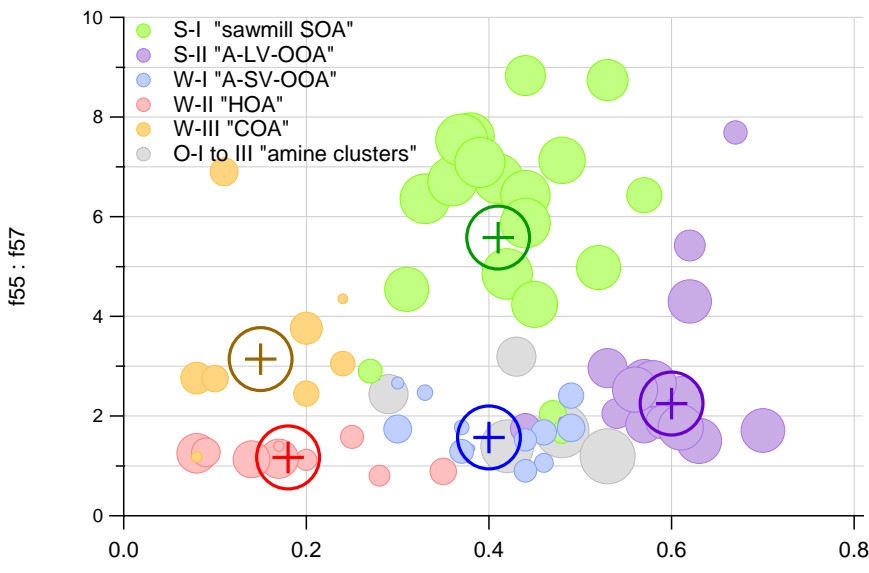

**Fig. 9.** "corr" $k = 8$ clustering solution projected onto 2-d axes, corresponding to *f44* derived oxidation level (estimated O:C; Aiken et al., 2008) and *f55:f57* ratio (truncated at 10) typically used for COA vs HOA source apportionment (Mohr et al., 2012). Marker size corresponds to silhouette value of the point, ranging from zero to one. Cluster centroid locations are marked separately with darker colours. Outlier clusters are shown in grey, without centroids.

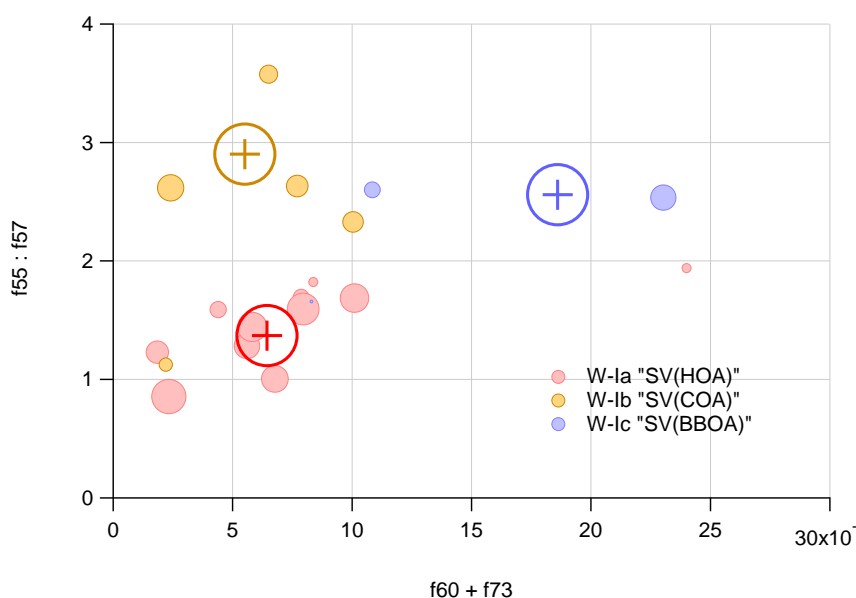

**Fig. 10.** Classification of A-SV-OOA (Cluster W-I) types into subgroups W-Ia, W-Ib and W-Ic, corresponding to SV(HOA), SV(COA) and SV(BBOA). 2-d visualisation is given in y-axis *f55/f57* (truncated to 4) and x-axis *f60+f73*. Marker size indicates object silhouette value. Other groups (W-II, W-III, S-X, O-X) are omitted from graph. Due to small sample size and low cohesion of clusters such a classification should be considered speculative at this point.



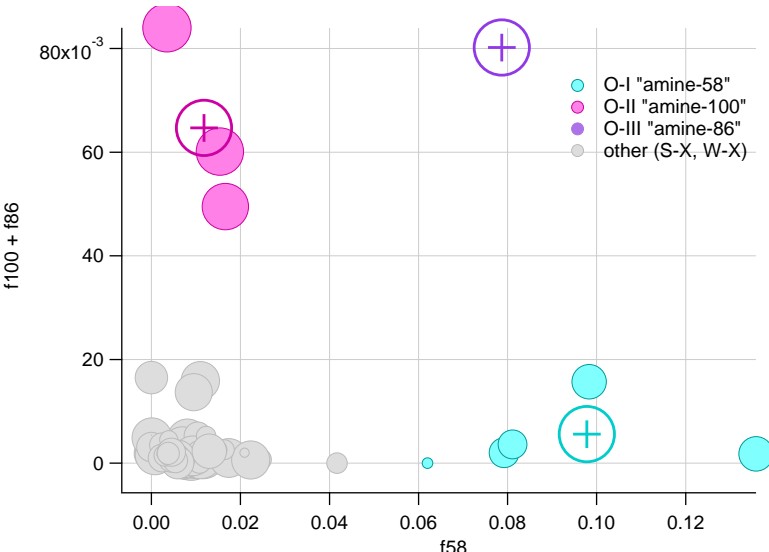

**Fig. 11. Outlier clusters (O-I to O-III) assumed to contain amines, and their respective centroids plotted in colour, and any other clusters in grey in *(f58, f86+f100)* spatial projection. Marker size corresponds to object silhouette value (unavailable for the singleton cluster O-III).**

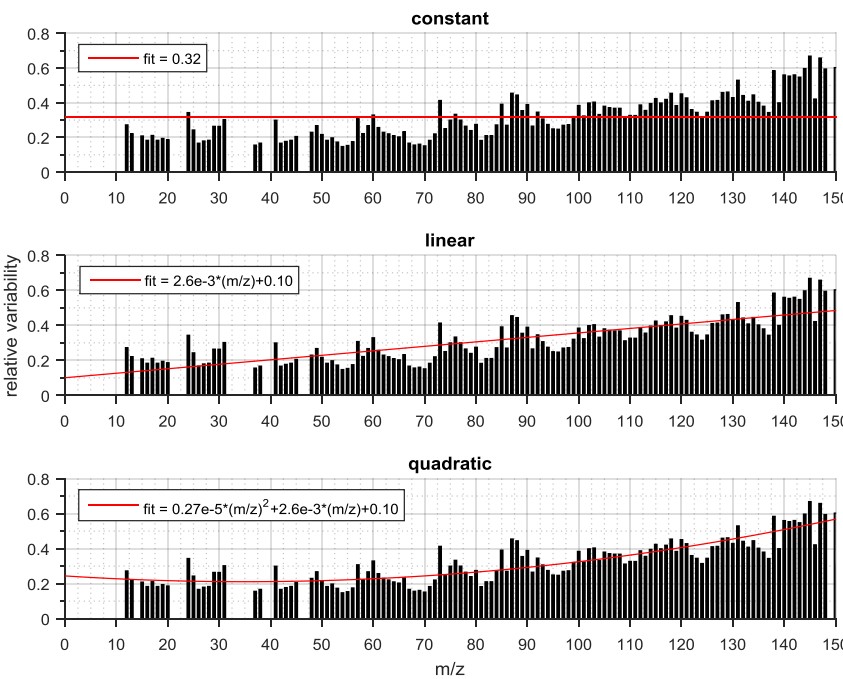

**Fig. 12. Silhouette-weighted standard deviation, as a function of *m/z*, for cluster S-I (sawmill-SOA). Upper panel: constant value ("a-value") estimate. Middle panel: linear regression estimate. Lower panel: exponential (quadratic) regression estimate.**





**Table 1.** Timeframes of measurements and numbers of successfully extracted pollution spectra per dataset.

| CAMPAIGN | START DATE | END DATE | # OF DAYS | SPECTRA EXTRACTED |
|---|---|---|---|---|
| "May 2008" | 29 Apr 2008 | 8 Jun 2008 | 40 | 23 |
| "September 2008" | 10 Sep 2008 | 15 Oct 2008 | 35 | 25 |
| "March 2009" | 4 Mar 2009 | 29 Mar 2009 | 25 | 33 |

**Table 2.** Diagnostics values, clustering parameters and cluster populations for solutions of 6 to 10 clusters. Oxidation level is described for each cluster centroid and potential sources are (preliminarily) identified. Within-cluster silhouette values are colour coded for readability (< 0.24 orange, 0.25…0.49 yellow, > 0.5 green). Solutions chosen for further analysis ("corr" k = 8 and k = 10) are highlighted in red.

| $s_m$ | k | TOTAL SOLUTION silh | S-I sawmill-SOA INTERMEDIATE OXIDISED n | silh | S-II mixed anthropogenic HIGHLY OXIDISED n | silh | W-I mixed anthropogenic INTERMEDIATE OXIDISED n | silh | W-Ia traffic (HOA) INTERMEDIATE OXIDISED n | silh | W-Ib cooking (COA) INTERMEDIATE OXIDISED n | silh | W-Ic biomass burning (BBOA) INTERMEDIATE OXIDISED n | silh | W-II traffic (HOA) FRESH n | silh | W-III cooking COA (+BBOA?) FRESH n | silh | O-I amine source? OUTLIER (mz 58) n | silh | O-II amine source? OUTLIER (mz 100, mz 86) n | silh | O-III amine source? OUTLIER (mz 86) n | silh |
|---|---|---|---|---|---|---|---|---|---|---|---|---|---|---|---|---|---|---|---|---|---|---|---|---|
| **sqEucl** | | | | | | | | | | | | | | | | | | | | | | | | |
| 1.32 | 6 | 0.46 | 23 | 0.55 | 17 | 0.47 | 23 | 0.38 | | | | | | | 10 | 0.38 | | | 5 | 0.47 | 3 | 0.77 | | |
| **Corr** | | | | | | | | | | | | | | | | | | | | | | | | |
| 1.62 | 6 | 0.45 | 23 | 0.66 | 17 | 0.51 | 21 | 0.30 | | | | | | | | | 11 | 0.33 | 6 | 0.15 | 3 | 0.90 | | |
| 1.23 | 7 | 0.46 | 23 | 0.65 | 17 | 0.57 | 15 | 0.25 | | | | | | | 9 | 0.41 | 8 | 0.35 | 6 | 0.08 | 3 | 0.86 | | |
| **1.21** | **8** | **0.48** | **23** | **0.65** | **17** | **0.56** | **15** | **0.25** | | | | | | | **9** | **0.41** | **8** | **0.35** | **5** | **0.48** | **3** | **0.86** | **1** | **N/A** |
| 1.42 | 9 | 0.46 | 21 | 0.65 | 16 | 0.53 | | | 12 | 0.34 | 7 | 0.32 | | | 10 | 0.33 | 6 | 0.16 | 5 | 0.49 | 3 | 0.89 | 1 | N/A |
| **1.36** | **10** | **0.46** | **21** | **0.64** | **16** | **0.54** | | | **11** | **0.35** | **6** | **0.28** | **3** | **0.17** | **9** | **0.37** | **6** | **0.23** | **5** | **0.49** | **3** | **0.88** | **1** | **N/A** |
| **Cos** | | | | | | | | | | | | | | | | | | | | | | | | |
| 1.66 | 6 | 0.45 | 23 | 0.63 | 17 | 0.49 | 20 | 0.31 | | | | | | | 12 | 0.38 | | | 6 | 0.13 | 3 | 0.88 | | |
| 1.69 | 7 | 0.47 | 23 | 0.63 | 17 | 0.47 | 20 | 0.32 | | | | | | | 12 | 0.38 | | | 5 | 0.50 | 3 | 0.88 | 1 | N/A |
| 1.15 | 8 | 0.48 | 23 | 0.63 | 17 | 0.54 | 13 | 0.38 | | | | | | | 9 | 0.35 | 10 | 0.24 | 5 | 0.47 | 3 | 0.82 | 1 | N/A |
| 0.96 | 9 | 0.45 | 21 | 0.64 | 16 | 0.54 | | | 13 | 0.23 | 8 | 0.32 | | | 9 | 0.34 | 5 | 0.25 | 5 | 0.44 | 3 | 0.79 | 1 | N/A |



**Table 3. Within-cluster, silhouette-weighted variabilities parametrised using constant, linear and exponential (quadratic) least squares regressions to the clustered data (variability [y] as a function of *m/z* [x]). The constant variability estimate corresponds to the "a-value" approach used in ME-2 analysis for constraining the PMF model.**

| CLUSTER | | CONSTANT $y = a$ | LINEAR FIT $y = bx + a$ | | QUADRATIC FIT $y = cx^2 + bx + a$ | | |
|---|---|---|---|---|---|---|---|
| | | a | b ($10^3$) | a | c ($10^5$) | b ($10^3$) | a |
| S-I | sawmill SOA | 0.32 | 2.56 | 0.10 | 2.70 | -1.89 | 0.24 |
| S-II | A-LV-OOA | 0.34 | 1.81 | 0.19 | 1.60 | -0.82 | 0.27 |
| W-I | A-SV-OOA | 0.18 | 0.59 | 0.13 | 1.28 | -1.51 | 0.19 |
| W-II | HOA | 0.41 | 1.60 | 0.28 | 5.37 | -7.25 | 0.56 |
| W-III | COA | 0.24 | -0.09 | 0.24 | 3.25 | -5.44 | 0.41 |