# Peer review of "Resolving anthropogenic aerosol pollution types - deconvolution and exploratory classification of pollution events"

_Atmospheric Chemistry and Physics, 2016_

## Referee Comment (RC1) · Anonymous Referee #1 · 21 Sep 2016

This paper presents the application of k-means clustering to AMS data recorded at Hyytialla. While techniques like PMF are more commonly used as a data reduction tool in the AMS community, clustering presents some interesting possibilities, specifically for the purposes of plume classification. The paper uses an interesting technique whereby PMF is used to screen out discrete plumes first and then clustering applied to the outputs, so that plumes can be grouped and examined free of the influence of background aerosols.

This work is relevant to ACP and there are a lot of good features in this manuscript, such as a thorough evaluation of different distance metrics and determination of the correct number of clusters, two things that are absolutely crucial and yet frequently

missed from some of the more naïve applications of clustering within atmospheric science. However, the paper is not without its weaknesses; it is written in a very rambling, overly conversational and at times woolly tone, which made it very difficult for me (as a reviewer) to get at the hard-and-fast science. Certain key details regarding the methodology are also not covered in sufficient detail. In short, I would say there is good science in here, but it does need more work to turn it into a good paper.

General:

This paper is very technical in nature and risks being outside of the scope of ACP and more suited to something like AMT (especially seeing as the authors seem to imply in a number of places that the technique needs more work). In order to remain in scope, I recommend that the abstract and conclusions contain more of an emphasis on the new insights to atmospheric science that this work has offered.

The language used in this paper is very conversational and more in the style of a magazine article or opinion piece, with the insertion of many words that serve no tangible purpose to the paper (a few are picked out in the technical comments). While this would mainly be considered cosmetic and probably not worth making too big a fuss over, the authors at times risk crossing the line to using 'weasel words', i.e. the insertion of adjectives that convey an opinion-based or otherwise unsubstantiated point to the reader. Examples of this would include describing the tools as "somewhat underused" on page 2 or the use of the CTOF as "advantageous" on page 4. This practice is to be discouraged in scientific literature, so I would suggest the authors generally revise the text to a more formal style, sticking to the facts as much as possible.

Also regarding presentation style, there is a tendency to start sections with a loquacious preamble describing previous work or setting out the need for a particular technique to be applied, however in a number of cases (see specific comments) this level of detail is completely unnecessary because many of these motivations are so well established in the literature they would be considered common knowledge in the com-

munity. While this too could be considered cosmetic, in places it seems that this writing is done at the expense of necessary technical detail. An example (described below) would be the extensive text devoted to comparing the CTOF with HRTOF (which could be handled by a decade-old citation) but insufficient detail on the unique features of the specific instrument used here.

Finally regarding writing style, the supplement is very rambling in its opening sections. While collating quotes from old textbooks would be a good primer for a non-specialist, I would tighten the text up a bit and focus more on what is specifically important for this work. I would also try to avoid repeating material that is already covered in the main article.

Contrary to what is frequently implied, this paper does not represent the first use of clustering applied to AMS data; people were trying it long before PMF was used, an example of which is the Marcolli et al. (2006) paper cited in both the manuscript and supplement (although it is incorrectly presented as an example of factorisation in the introduction). As a technique for analysing ambient data, clustering failed to gain traction within the AMS community (reflected in the low number of publications) because unlike SPMS, AMS mass spectra do not (generally) represent discrete events so therefore interpreting clustering outputs carries with it many inherent problems. While this paper addresses many of these limitations, the authors would do well to tone down much of the text (in particular in the introduction) that seems to work off the principle that the application of clustering to AMS data is completely new. The real novelty of this work is the combination of clustering with discrete PMF analysis to get at data from specific plumes, which should be better suited to clustering than the blanket application to all recorded spectra, so I would spend more time focusing on this aspect of the work when demonstrating novelty.

A general fundamental weakness with clustering as applied to AMS data, even as applied here, is that it is not capable of identifying individual components when a measured mass spectrum is composed of an indeterminate combination of different com-

ponents, rather than a single type. While this would not be the case here if individual plumes could be attributable to single point sources, it would not be true of plumes from mixed sources, which may occur with urban plumes consisting of a mixture of traffic, cooking, etc. However, this very fundamental limitation is not really discussed, in particular in section 3.4.2, where the analysis appears to have been approached from the perspective that discrete clusters representing these types should be expected. I would argue that good clustering for these potentially overlapping sources should not be expected and the fact that these are represented by 'weak' clusters should come as no surprise. It is also completely overlooked when suggesting that the variability within clusters could inform the a-values used in ME-2. The text should really take this whole issue into account better. I would note that the use of a fuzzy clustering algorithm (e.g. c-means) may at least partially overcome this issue, but this presents an entirely new avenue of work outside of the scope of this paper.

Specific comments:

Title: The title of the paper is very obtuse and says very little about the actual content. Recommend rewording as something that includes the words 'clustering' and 'plumes'.

The first two paragraphs of the introduction are a little superfluous considering how well established mass spectrometry and the need for associated data mining and reduction is within atmospheric science. Given that there is a wealth of articles already published in ACP on mass spectrometric data reduction, I would remove this text.

Pages 3-4: It is difficult to sell the CTOF as advantageous given that the more diverse variable set provided by the HRTOF would almost certainly lead to a better statistical treatment (this is pretty much said later in the manuscript). But then the authors shouldn't have to justify using a CTOF over a HRTOF because the choice of the specific pre-existing dataset is justified later in the paper and the focus of the paper is on the analysis technique anyway. More generally, given how well established both instruments are, it is really not necessary to describe the mass spectrometry technology in

this much detail; a simple citation of the literature would suffice.

Page 4: Regarding the differences between this instrument and a standard CTOF, is this described elsewhere in the literature? If so, these should be cited. If not, much more detail should be given here, particular as regards the helium bleed system, ideally with a technical schematic.

Page 5: A description of the diagnostic that lead the authors to be concerned of the airbeam linearity would be appropriate. Was the airbeam affected by the helium feature of the instrument?

Page 8: The criteria given for plume identification are very qualitative and therefore subjective. Can some quantities be assigned to any of the criteria, such as rise rate or duration? These would contribute to the general goal of a truly objective system of data reduction, even if it is not achieved here.

Page 8: The justification for using k-means seems a little overwrought. To be clear, k-means is not the most simple algorithm in existence (hierarchical agglomerative clustering can probably claim that), but it is nevertheless generally treated as the 'default go-to' algorithm by most people in absence of a reason to use anything else because of its simplicity of operation and low computational cost. It's difficult to see this being any different in this case, so it would be better to simply state that you chose k-means for this reason and that a comparison with other algorithms could be done as future work.

Page 10, line 11: Saying that rotational ambiguity is 'mostly avoided' is a strong statement. What evidence can the authors present to back this up?

Page 12: Why not use the same weighting function as the error model used to weight the PMF residuals?

Page 14, line 21: "We hope" is a very odd thing to say. Can the robustness of the method not be tested somehow?

Page 20: After all the discussion regarding the selection of the correct distance metrics for mass spectra during clustering (particularly in the supplement), why use Pearson's R here?

Page 24: Referring to a fundamental limitation of clustering as a technique (see general comments), the authors should take account of the fact that some plumes may hypothetically consist of a mixture of individual sources.

Section 3.4.3 seems rather long and tangential considering that it fails to reach a definitive conclusion. Given that this is by no means the first time amines have been reported at Hyytialla, I would shorten this section for the sake of brevity.

Page 29: Again, the authors fail to acknowledge that the within-cluster variability can be caused by the varying influences of different sources within mixed plumes. Following from this, the later statements that "…the variabilities implied by this study can be used as an indicator of what the likely magnitude of the underlying natural variability within the observed classes of aerosols…" and "…the natural variability within an aerosol type may be significantly larger than what is often allowed in conjunction with the constrained PMF/ME-2." should have the caveats added that this will only work if the plumes can be absolutely verified as being of a single source.

Page 29: A frequently-used tool for quantifying rotational ambiguity is the PFEAK parameter in PMF, yet this is not even mentioned. Why was this not used? This would seem particularly appropriate here because when looking at 2-factor solutions, the limitations of applying a global parameter to explore the solution space are significantly reduced.

Page 31: The comparison with PMF in the conclusions is extremely disingenuous because the authors fail to distinguish between the two very different data models employed by the different algorithms and the very different way in which they can be used. It also seems strange to compare these like this because the clustering technique used here relies on PMF to extract the plumes in the first instance. While clustering is good

at analysing discrete plumes, its data model cannot handle arbitrary mixtures, which can make up the majority of AMS data in many cases. It is wrong to say that clustering eliminates the problem of rotation (there may still be some rotation in the plume extraction part – the authors have not discounted this) or that the cluster variabilities can be used to estimate source variability (it will only work for pure-component plumes). I recommend reworking this section to focus on what scientific insights this offers in addition to PMF, rather than pitching the two against each other.

Technical corrections:

Page 3, line 21: What is so "regrettable" about a full review of the statistical techniques being out of scope? As a reviewer of an atmospheric science paper, I confess I was actually quite relieved it wasn't in there.

Page 4, line 9: The word "specimen" is a very peculiar choice and not necessary. Please remove.

Page 4: The need to process AMS data correctly is very well established in the literature, so the opening text of section 2.1.2 is unnecessary.

Page 8, line 22: "…to thoroughly evaluate events' satisfaction our selection criteria" makes no sense. Please revise.

Page 9, line 8: "Achilles heel" is inappropriate language, seeing as it is an inherent feature of the data model applied and associated constraints, not PMF specifically. Please remove.

Page 12, line 1: Remove "unfortunately"

Page 13, line 28: Remove "with its own unavoidable weaknesses". It's not possible to make this statement without an algorithm in mind.

Page 14, line 4: Remove the word "Obviously". It would not be obvious to a reader with no experience of this.

Page 20, line 6: Remove "on the other hand" but also generally check the wording of the sentence; I'm not 100% sure what it is that is being said.

Page 21, line 20: The use of the future tense in "We will call. . ." is again overly conversational in tone.

Page 22, line 19: Please be specific when referring to "the last hypothesis". I had to read this several times before I thought I understood it.

Figure 1: This needs extensive tidying up, specifically to avoid lines overlaying the axis labels of inset graphs. Also, I would use legends rather than referring to colours in the text.

Supplement: This would be much easier to follow if the figures and tables were presented alongside the associated text.

Page S4, line 15: Should be 'in practice' ('practise' is the verb form in UK English)

―――――――――――――――――――――

---

## Referee Comment (RC2) · Anonymous Referee #2 · 11 Oct 2016

This manuscript describes an interesting application of cluster analysis for analysis of ambient aerosol data obtained with an Aerosol Mass Spectrometer. In this method, short pollution time periods are analyzed with positive matrix factorization. The factorization yields background and pollution factor mass spectra that are then analyzed with cluster analysis to classify the distinct types of pollution factors that are obtained. As currently written, this manuscript introduces technical details of an analysis method and would seem to be more appropriate for AMT than ACP. It is important for the authors to highlight how this technique provides improved or new insight into study of atmospheric aerosols so that inclusion in ACP is better justified. I recommend publication in ACP after this change and changes suggested below are made:

[Figure]

Main Comments

1) In general, the paper is a little longer than it needs to be because it includes a lot of detailed background information about some topics while not enough information is given about necessary details. For example, page 8, Lines 10-13 only offer no quantitative information about how air pollution events are selected. Can some of the words such as "temporary", "distinct rise", and "unambiguous separation of pollution plume from background" be quantified?. 2) In figure 1, pollution events of varying time scales and multiple apparent pollution peaks are seen. What exactly is the process used to make these selections? What controls the length of the time period that is used as a pollution event? Is wind direction data used for selection? What is the sensitivity of the PMF pollution event solutions to the exact time period range selected around the pollution event? Did you consider as an alternative to this manual plume method to run a traditional PMF analysis on the entire dataset and identify plumes as time periods where the residuals of the PMF analysis are high? 3) It is not clear to me why the pollution event PMF analysis used in this manuscript necessarily provides a more unambiguous separation of "pollution" and "background" than PMF analyses that are performed on the whole dataset. In fact, if the pollution event is simply a result of changes in wind direction that mixes in a different well mixed airmasses, then the PMF factor that is extracted would necessarily be just an average mass spectrum of all the sources present in the polluting airmass. No advantage would have been gained by this method to allow separation of individual sources and this would seem to be a weakness of this method. This aspect is not discussed in the manuscript. 4) The manuscript refers to ambiguities in PMF analysis as a weakness and implies that this analysis somehow solves or provides a better solution to this problem of ambiguity. In fact, the manuscript clearly states the difficulty of separating the various primary aerosol sources. One of the advantages of the traditional method of doing PMF or ME-2 on the entire dataset in this context could be the fact that it can exploit differences in temporal profiles of primary sources (i.e. different diurnal cycles) and also exploit the fact that source mass spectra are similar to allow for separation of multiple primary

sources within a well mixed pollution event (i.e. a event such as that mentioned in comment 3 above). A comparison between the classification results and a traditional PMF of the entire dataset would have been a good way to address this and to highlight similarities and differences in results. The manuscript should more clearly state discuss the advantages/disadvantages of using this method compared to PMF. 5) The strongest part of this manuscript is the application and interpretation of the various clustering metrics to understand similarity and differences between the cluster spectra. It may be useful to highlight more strongly how these metrics could be applied to spectra obtained with typical PMF/ME-2 analysis. Would use of the cluster analysis metrics to reference spectra and PMF solutions provide a means of automating classification in PMF analysis? Also an intriguing part of this that could be discussed in more detail is the possibility to use the cluster analysis to define a-values and reference spectra for ME-2.

---

## Author Comment (AC1) · 29 Nov 2016

This paper presents the application of k-means clustering to AMS data recorded at Hyytialla. While techniques like PMF are more commonly used as a data reduction tool in the AMS community, clustering presents some interesting possibilities, specifically for the purposes of plume classification. The paper uses an interesting technique whereby PMF is used to screen out discrete plumes first and then clustering applied to the outputs, so that plumes can be grouped and examined free of the influence of background aerosols.

This work is relevant to ACP and there are a lot of good features in this manuscript, such as a thorough evaluation of different distance metrics and determination of the correct number of clusters, two things that are absolutely crucial and yet frequently missed from some of the more naïve applications of clustering within atmospheric science. However, the paper is not without its weaknesses; it is written in a very rambling, overly conversational and at times woolly tone, which made it very difficult for me (as a reviewer) to get at the hard-and-fast science. Certain key details regarding the methodology are also not covered in sufficient detail. In short, I would say there is good science in here, but it does need more work to turn it into a good paper.

We would like to thank the referee for the comments and especially the constructive suggestions to improve manuscript readability. Based on the comments, we

- modified the general text style towards a more compact presentation, and generally revised wordings to conform to a more formal writing style.
- added and/or improved description regarding the methodology-related questions raised by the referee

Please find more detailed responses to the comments and questions in the following:

General:

This paper is very technical in nature and risks being outside of the scope of ACP and more suited to something like AMT (especially seeing as the authors seem to imply in a number of places that the technique needs more work). In order to remain in scope, I recommend that the abstract and conclusions contain more of an emphasis on the new insights to atmospheric science that this work has offered.

Due to the focus of the paper being divided between (1) chemometrics related data analysis and (2) aerosol chemical interpretation of results, both of which are necessary to demonstrate the usefulness of our approach, we tried to balance between the two viewpoints. However, in accordance with the referee's suggestions we have shifted the focus of the manuscript more towards the discussion of the phenomenon of air pollution plumes.

The beginning of introduction was rewritten and emphasis on the anthropogenic pollution was added. In conclusions similar shift of emphasis was done.

The language used in this paper is very conversational and more in the style of a magazine article or opinion piece, with the insertion of many words that serve no tangible purpose to the paper (a few are picked out in the technical comments). While this would mainly be considered cosmetic and probably not worth making too big a fuss over, the authors at times risk crossing the line to using 'weasel words', i.e. the insertion of adjectives that convey an opinion-based or otherwise unsubstantiated point to the reader. Examples of this would include describing the tools as "somewhat underused" on page 2 or the use of the CTOF as "advantageous" on page 4. This practice is to be discouraged in scientific literature, so I would suggest the authors generally revise the text to a more formal style, sticking to the facts as much as possible.

As suggested, we have revised much of the expressions to comply with a more formal register.

Also regarding presentation style, there is a tendency to start sections with a loquacious preamble describing previous work or setting out the need for a particular technique to be applied, however in a number of cases (see specific comments) this level of detail is completely unnecessary because many of these motivations are so well established in the literature they would be considered common knowledge in the community. While this too could be

considered cosmetic, in places it seems that this writing is done at the expense of necessary technical detail. An example (described below) would be the extensive text devoted to comparing the CTOF with HRTOF (which could be handled by a decade-old citation) but insufficient detail on the unique features of the specific instrument used here.

We attempted to accommodate a larger non-specialist readership, and due to the rather wide array of topics covered, the "community" becomes equally broad. We try to cater for readers interested in such data analysis methodology in general as well as aerosol chemists focusing on the mass spectral results.

However, we have reviewed these cases and omitted/shortened preambles, as well as rewrote the said CToF vs HRToF part.

Finally regarding writing style, the supplement is very rambling in its opening sections. While collating quotes from old textbooks would be a good primer for a non-specialist, I would tighten the text up a bit and focus more on what is specifically important for this work. I would also try to avoid repeating material that is already covered in the main article.

In the supplement we aimed to provide some background information for a non-specialist reader, as we assumed it would be beneficial to the fraction of readers less familiar with statistical methods such as PMF or clustering. We would prefer to keep it this way, but have added a table of content to the beginning of the SI in order to make it easier for a reader to find the specific details they came looking for from the SI.

Contrary to what is frequently implied, this paper does not represent the first use of clustering applied to AMS data; people were trying it long before PMF was used, an example of which is the Marcolli et al. (2006) paper cited in both the manuscript and supplement (although it is incorrectly presented as an example of factorisation in the introduction). As a technique for analysing ambient data, clustering failed to gain traction within the AMS community (reflected in the low number of publications) because unlike SPMS, AMS mass spectra do not (generally) represent discrete events so therefore interpreting clustering outputs carries with it many inherent problems. While this paper addresses many of these limitations, the authors would do well to tone down much of the text (in particular in the introduction) that seems to work off the principle that the application of clustering to AMS data is completely new. The real novelty of this work is the combination of clustering with discrete PMF analysis to get at data from specific plumes, which should be better suited to clustering than the blanket application to all recorded spectra, so I would spend more time focusing on this aspect of the work when demonstrating novelty.

We thank the referee for pointing out this chance of misinterpretation, as we did not want to imply that we are the first to apply clustering methods to AMS data, and have clarified this part of the introduction and conclusions. What we do aspire to affirm within this work is that clustering can succesfully be used for the classification of discrete AMS sample spectra, whether deconvolved from ambient observations or laboratory, which has not to our knowledge been shown outside the specific niche of single particle applications.

Having Marcolli et al., in the references for early work factor analytical techniques is indeed an error – it is used as a dimensionality reductive technique yes, but not "factorisation". This sentence has now been corrected, and a separate mention added on the Marcolli et al. reference.

A general fundamental weakness with clustering as applied to AMS data, even as applied here, is that it is not capable of identifying individual components when a measured mass spectrum is composed of an indeterminate combination of different components, rather than a single type. While this would not be the case here if individual plumes could be attributable to single point sources, it would not be true of plumes from mixed sources, which may occur with urban plumes consisting of a mixture of traffic, cooking, etc. However, this very fundamental limitation is not really discussed, in particular in section 3.4.2, where the analysis appears to have been approached from the perspective that discrete clusters representing these types should be expected. I would argue that good clustering for these potentially overlapping sources should not be expected and the fact that these are represented by 'weak' clusters should come as no surprise. It is also completely overlooked when suggesting that the variability withinclusters could inform the a-values used in ME-2. The text should really take this whole issue into account better. I would note that the use of a fuzzy clustering algorithm (e.g. c-means) may at least partially overcome this issue, but this presents an entirely new avenue of work outside of the scope of this paper.

We want to thank the referee for pointing out that discussion on this issue is missing from the manuscript. It has been added to Sections 2.3.1 and 3.4. We also modified Sect. 3.6. by adding this additional mention regarding the effect on ME-2 variability estimate, although we did already touch upon this issue, stating that the inter-cluster variability should be considered an upper limit of the actual variability.

We fully agree with fuzzy clustering algorithms being perhaps the most promising way to go in future classification of ambient aerosol types. For brevity we omitted this discussion since an earlier version of the manuscript, but will restore a short recommendation to the conclusions.

Specific comments:

Title: The title of the paper is very obtuse and says very little about the actual content. Recommend rewording as something that includes the words 'clustering' and 'plumes'. The first two paragraphs of the introduction are a little superfluous considering how well established mass spectrometry and the need for associated data mining and reduction is within atmospheric science. Given that there is a wealth of articles already published in ACP on mass spectrometric data reduction, I would remove this text.

Since the main message and much of the discussion in this work is to encourage the use of chemometric, machine driven analysis methods as a source of chemical information in general (and in connection with AMS especially), and we consider the exact methods/algorithms chosen to be "of secondary interest", we'd prefer not to clutch on to the single technical method (clustering, PMF, mass scaling, dissimilarity metric etc) in the title.

To keep better in line with this thought, but to make the title less obtuse, we reformulated the title to "Resolving anthropogenic aerosol pollution chemotypes - deconvolution and exploratory classification of pollution events".

As suggested, the two first paragraphs of the introduction were omitted and the next paragraph edited accordingly.

Pages 3-4: It is difficult to sell the CTOF as advantageous given that the more diverse variable set provided by the HRTOF would almost certainly lead to a better statistical treatment (this is pretty much said later in the manuscript). But then the authors shouldn't have to justify using a CTOF over a HRTOF because the choice of the specific pre-existing dataset is justified later in the paper and the focus of the paper is on the analysis technique anyway. More generally, given how well established both instruments are, it is really not necessary to describe the mass spectrometry technology in this much detail; a simple citation of the literature would suffice.

We shortened and reformulated this paragraph according to the referee's comments..

Page 4: Regarding the differences between this instrument and a standard CTOF, is this described elsewhere in the literature? If so, these should be cited. If not, much more detail should be given here, particular as regards the helium bleed system, ideally with a technical schematic.

The modification is described very briefly in several earlier publications (e.g. Corrigan et al., 2013), but not in an extensive way that would add to the information given here. We have extended and clarified this description in the manuscript. Importantly, we now clearly note that He is simply bled into the ptof chamber to increase the overall pressure in this region. We realise that our original text could be interpreted that the He flow was focused specifically at the sample beam.

Page 5: A description of the diagnostic that lead the authors to be concerned of the airbeam linearity would be appropriate. Was the airbeam affected by the helium feature of the instrument?

The discrepancy is apparent e.g. from observing the $Ar/N_2/O_2$ signal ratios seen by the instrument not matching the molecular abundances present in the atmosphere. This information was added. The airbeam was decreased due to the He added, and the measured composition ($O_2/N_2/Ar$) was brought closer to real atmospheric values by the Helium feature, but was still considered prone to errors upon changes in detector gain etc. Discussion was added to text along with a reference (Hings et al., 2007).

Page 8: The criteria given for plume identification are very qualitative and therefore subjective. Can some quantities be assigned to any of the criteria, such as rise rate or duration? These would contribute to the general goal of a truly objective system of data reduction, even if it is not achieved here.

The referee is quite right about the subjectivity and qualitative nature of the selection. We did not have exact, quantitative thresholds for accepting or rejecting an air pollution event or a data reduction solution. Some quantitative limits could be assigned, but to be less subjective they should ideally be based on something, such as some statistical parameters of pollution plumes or literature values for typical events etc. Without the benefit of such prior information or previous analyses available, we opted for this manual, qualitative approach.

To give the reader and referee some idea on the magnitude of the phenomena described qualitatively, we have tried to estimate some values for the uncertainty of our selections, and have added this to Sect. 2.2.2. Unfortunately our chemometrics-oriented analysis does not enable us to give any specific guidelines for formulating an objective (or automatic) air pollution identification system. We agree such a system would be highly desirable.

Page 8: The justification for using k-means seems a little overwrought. To be clear, k-means is not the most simple algorithm in existence (hierarchical agglomerative clustering can probably claim that), but it is nevertheless generally treated as the 'default go-to' algorithm by most people in absence of a reason to use anything else because of its simplicity of operation and low computational cost. It's difficult to see this being any different in this case, so it would be better to simply state that you chose k-means for this reason and that a comparison with other algorithms could be done as future work.

Since the selection of statistical methods befitting the purpose of AMS spectra classification is an integral part of this work, we wanted to emphasise the fact that the selection of clustering algorithm is not a trivial task, but something that can and should be given thought and subjected to critique in similar future work. Our going with "the default" algorithm and not evaluating the options can therefore be seen as an oversight by some – here we want to acknowledge and emphasise the fact that the selection of k-means is only a first try, and is not based on any performance evaluation unlike most of the other technical choices made in this study. We have reformulated the paragraph to hopefully convey this message better.

Page 10, line 11: Saying that rotational ambiguity is 'mostly avoided' is a strong statement. What evidence can the authors present to back this up?

As Paatero and Hopke (2009) write in their introduction, the problem of selecting from among available rotations can be reduced if there is additional information available on the system in question:

> "For many chemical systems, there is information available as to the nature of the underlying patterns in the resolved components. […] There may be known spectral features such as regions of zero absorbance that can be used to reduce the rotational space."

In our case similar, albeit physical, information is provided by the (qualitative) knowledge on the temporal behaviour of aerosol mass concentrations in mixing of air pollution plume and the background aerosol. Also, applying the PMF analysis to pollution events, with a plume surrounded by background regions, introduces the "control regions" points with (near) zero concentrations of the pollutants. Having the many "zero values" considerably reduces and may even eliminate rotational ambiguity (Paatero et al., 2014; Anderson, 1984).

We concede this qualitative way of evaluating and selecting the solutions (a posteriori) is far from the ideal, and the same should optimally be implemented quantitatively within the PMF model resolving via e.g. pulling/constraining the time series in the iterative, solving process itself. However, such functionality would require modification to the user interface we had available, as restricting the time series (a priori) in the current framework (SoFi U.I.) would require an anchor for F (i.e. reference time series) which we don't have available beforehand.

Page 12: Why not use the same weighting function as the error model used to weight the PMF residuals?

This is an interesting idea, but one we did not test as an alternative. The approach is philosophically different from the justification of testing the mass/intensity weighting, which is to weight based on information value available in specific masses (e.g. high vs low m/z, deriving from information loss in fragmentation), whereas the PMF weighting model is (mostly) based on experiment-derived estimates for measurement uncertainty. Also, we are unsure if the same error model would apply to PMF output as input, as the model has tried to take into account the uncertainty already. Perhaps rather some methods should be used where the PMF output uncertainty is estimated specifically?

We agree this approach would warrant investigation in future work, and in case of a positive outcome there is no reason why such a weighting model could/should not combine well with the afore mentioned "mass scaling".

Page 14, line 21: "We hope" is a very odd thing to say. Can the robustness of the method not be tested somehow?

We agree, and rephrased the sentence. However, we lack tools to evaluate how fruitful this approach is. But it's the best we could come up with, to alleviate the concern of outliers overly affecting the centroids. At least it is logical and solidly statistics based. We would be happy to be pointed towards a better method, but in the longer run we would suggest choosing a clustering algorithm without these inherent weaknesses of hard classification and susceptibility to outliers, which would eliminate the need for such posteriori weighting altogether. In absence of previous literature or e.g. a synthetic data set to analyse, the effectivity is difficult to evaluate conclusively. We note that in the end the effect on the centroids is really small, as per Figure 6 and the correlation of $r_s^2$ of 0.994 between the scaled and unscaled spectra.

Page 20: After all the discussion regarding the selection of the correct distance metrics for mass spectra during clustering (particularly in the supplement), why use Pearson's R here?

Because we experimentally found it to also perform best out of the similarity metrics tested. This was clarified in the text.

Page 24: Referring to a fundamental limitation of clustering as a technique (see general comments), the authors should take account of the fact that some plumes may hypothetically consist of a mixture of individual sources.

We have added the disclaimer also here. Please see answers to the other related comments. It should be noted clustering as a technique is not at fault if it is used on a data set which does not comply with underlying assumptions (for hard classification data is not discrete, if some objects in fact belong to multiple classes or between them).

Section 3.4.3 seems rather long and tangential considering that it fails to reach a definitive conclusion. Given that this is by no means the first time amines have been reported at Hyytialla, I would shorten this section for the sake of brevity.

As suggested, we have revised this section, moving the diverging discussion into the supplement. However, as per the references we found and cited, amines have not been conclusively "reported" in biogenic background aerosol spectra or in Hyytiälä – we would say the mass spectral evidence is circumstantial at best, and we would rank our observations among the most direct ones. This is also one of the sections with new, atmospheric chemistry relevant findings of the study, and discussion on the possible origins and nature of biogenic amines should is one that we'd expect to especially interest ACP readers, (perhaps more than the methodology itself).

Page 29: Again, the authors fail to acknowledge that the within-cluster variability can be caused by the varying influences of different sources within mixed plumes. Following from this, the later statements that ". . .the variabilities implied by this study can be used as an indicator of what the likely magnitude of the underlying natural variability within the observed classes of aerosols. . ." and ". . .the natural variability within an aerosol type may be significantly larger than what is often allowed in conjunction with the constrained PMF/ME-2." should have the caveats added that this will only work if the plumes can be absolutely verified as being of a single source.

We agree with that the concern for the mixing phenomenon needs to be mentioned, and have added the disclaimer and discussion in this section as well. (Please see earlier comment on possible plume mixing and our answer). However, we do think the concern is alleviated by the similarity of within-cluster variabilities for the well-resolved clusters and the ones more prone to mixed plumes, as well as the post-processing in place to down-weight such effect.

Page 29: A frequently-used tool for quantifying rotational ambiguity is the PFEAK parameter in PMF, yet this is not even mentioned. Why was this not used? This would seem particularly appropriate here because when looking at 2-factor solutions, the limitations of applying a global parameter to explore the solution space are significantly reduced.

We omitted the Fpeak discussion from the manuscript as it was considered too much of a minor technical detail to be included. A consensus in the AMS PMF community seems to be that liberal use of Fpeak is to be discouraged due to the difficulty of producing yet more solutions (of mathematically inferior quality; higher Q value) for the analyst to select from, further complicating the solution selection process. The referee is correct that in our specific case is a "special application" where it should be better justified, due to the different solution selection/validation criteria.

We did scan non-zero Fpeak values where Fpeak=0 failed to produce an acceptable solution. In 7 cases (included in the total of 81 spectra) an acceptable rotation was found this way. The criteria for PMF solution acceptance remained the same, and when non-zero Fpeak was applied, extra care was taken not to allow factors with profiles that exhibit "unrealistic/unphysical behaviour" such as only containing noise-like spectra or no contribution to m/z 43 or 44 Th.

We added this information to the supplementary material (Sect. S.1).

Page 31: The comparison with PMF in the conclusions is extremely disingenuous because the authors fail to distinguish between the two very different data models employed by the different algorithms and the very different way in which they can be used. It also seems strange to compare these like this because the clustering technique used here relies on PMF to extract the plumes in the first instance. While clustering is good at analysing discrete plumes, its data model cannot handle arbitrary mixtures, which can make up the majority of AMS data in many cases. It is wrong to say that clustering eliminates the problem of rotation (there may still be some rotation in the plume extraction part – the authors have not discounted this) or that the cluster variabilities can be used to estimate source variability (it will only work for pure-component plumes). I recommend reworking this section to focus on what scientific insights this offers in addition to PMF, rather than pitching the two against each other.

It was not our intention to make it sound like we are pitching our approach against classical PMF analyses, and have rewritten large parts of this section to highlight the added value of our method. We do not state that clustering eliminates rotational problems in PMF (which it indeed does not).

Technical corrections:

Page 3, line 21: What is so "regrettable" about a full review of the statistical techniques being out of scope? As a reviewer of an atmospheric science paper, I confess I was actually quite relieved it wasn't in there.

We omitted the word "regrettable".

Page 4, line 9: The word "specimen" is a very peculiar choice and not necessary. Please remove.

Removed.

Page 4: The need to process AMS data correctly is very well established in the literature, so the opening text of section 2.1.2 is unnecessary.

We removed the introducing remarks.

Page 8, line 22: ". . .to thoroughly evaluate events' satisfaction our selection criteria" makes no sense. Please revise.

Should have been "evaluate events' satisfaction **of** our selection criteria". Was revised to "evaluate which of the events satisfy our selection criteria."

Page 9, line 8: "Achilles heel" is inappropriate language, seeing as it is an inherent feature of the data model applied and associated constraints, not PMF specifically. Please remove.

We have revised the expression to "factor analytical models' inherent weakness of rotational ambiguity, which also afflicts PMF (Paatero et al, 2014)"

Page 12, line 1: Remove "unfortunately"

Removed.

Page 13, line 28: Remove "with its own unavoidable weaknesses". It's not possible to make this statement without an algorithm in mind.

Removed.

Page 14, line 4: Remove the word "Obviously". It would not be obvious to a reader with no experience of this.

Reformulated.

Page 20, line 6: Remove "on the other hand" but also generally check the wording of the sentence; I'm not 100% sure what it is that is being said.

The sentence was revised.

Page 21, line 20: The use of the future tense in "We will call. . ." is again overly conversational in tone.

Changed to "we name".

Page 22, line 19: Please be specific when referring to "the last hypothesis". I had to read this several times before I thought I understood it.

The paragraph was revised in connection to earlier comment, the reference should be clearer now.

Figure 1: This needs extensive tidying up, specifically to avoid lines overlaying the axis labels of inset graphs. Also, I would use legends rather than referring to colours in the text.

Figure 1 graphics were revised.

Supplement: This would be much easier to follow if the figures and tables were presented alongside the associated text.

Good suggestion. We changed this.

Page S4, line 15: Should be 'in practice' ('practise' is the verb form in UK English)

Thank you. Changed.
* * *
**references:**

Anderson, T. W.: An Introduction to Multivariate Statistical Analysis, 2nd Edition, Wiley, New York, 1984.

Corrigan, A., Russell, L., Takahama, S., Äijälä, M., Ehn, M., Junninen, H., Rinne, J., Petäjä, T., Kulmala, M., and Vogel, A.: Biogenic and biomass burning organic aerosol in a boreal forest at Hyytiälä, Finland, during HUMPPA-COPEC 2010, *Atmospheric Chemistry and Physics*, 13, 12233-12256, 2013.

Hings, S. S., Walter, S., Schneider, J., Borrmann, S., and Drewnick, F.: Comparison of a quadrupole and a time-of-flight aerosol mass spectrometer during the Feldberg aerosol characterization experiment 2004. *Aerosol Science and Technology*, *41*, 679-691, 2007.

Marcolli, C., Canagaratna, M., Worsnop, D., Bahreini, R., De Gouw, J., Warneke, C., Goldan, P., Kuster, W., Williams, E., and Lerner, B.: Cluster analysis of the organic peaks in bulk mass spectra obtained during the 2002 New England Air Quality Study with an Aerodyne aerosol mass spectrometer, *Atmospheric Chemistry and Physics*, 6, 5649-5666, 2006.

Paatero, P., and Hopke, P. K.: Rotational tools for factor analytic models, *Journal of Chemometrics*, 23, 91-100, 2009.

Paatero, P., Eberly, S., Brown, S., and Norris, G.: Methods for estimating uncertainty in factor analytic solutions, *Atmospheric Measurement Techniques*, 7, 781-797, 2014.

---

## Author Comment (AC2) · 29 Nov 2016

This manuscript describes an interesting application of cluster analysis for analysis of ambient aerosol data obtained with an Aerosol Mass Spectrometer. In this method, short pollution time periods are analyzed with positive matrix factorization. The factorization yields background and pollution factor mass spectra that are then analyzed with cluster analysis to classify the distinct types of pollution factors that are obtained. As currently written, this manuscript introduces technical details of an analysis method and would seem to be more appropriate for AMT than ACP. It is important for the authors to highlight how this technique provides improved or new insight into study of atmospheric aerosols so that inclusion in ACP is better justified. I recommend publication in ACP after this change and changes suggested below are made:

We thank the referee for his/her valuable comments.

In accordance with similar comments by referee 1, the manuscript title was modified, large parts of the abstract, introduction, and conclusions were rewritten, and emphasis on the aerosol chemical conclusions was added.

Main Comments

1) In general, the paper is a little longer than it needs to be because it includes a lot of detailed background information about some topics while not enough information is given about necessary details. For example, page 8, Lines 10-13 only offer no quantitative information about how air pollution events are selected. Can some of the words such as "temporary", "distinct rise", and "unambiguous separation of pollution plume from background" be quantified?

We have attempted to implement most of the changes proposed to include all necessary details.

We have added the estimated thresholds used in the manual event selection along with more detailed descriptions for our selection criteria in Sect. 2.2.2. However, since the examination was only by visual inspection, and not a mathematical one, the limits are approximates.

We agree with the referee on that exact, quantitative criteria for pollution event selection would certainly be desirable and preferential to the approximated thresholds we used. This would eliminate on of the final sources of subjective judgement of this work. However, in this study, the selection was based on the said criteria due to our difficulty of properly evaluating the fulfillment of especially criteria 2 and 3 (page 8) on an exact level.

2) In figure 1, pollution events of varying time scales and multiple apparent pollution peaks are seen. What exactly is the process used to make these selections?

This is covered in previous comment/answer (#1). The pollution episodes are selected based on the criteria mentioned above. The selection process description is now expanded on in the text.

What controls the length of the time period that is used as a pollution event?

The time of "increased OA concentration", as verified by a PMF factor emerging and disappearing in the spectra extraction analysis.

Is wind direction data used for selection?

Wind data is not used in selection of pollution events, as there are events that could arise from momentary emissions without a change in wind direction (e.g. passing vehicles, cooking, igniting a fire at a fireplace).

What is the sensitivity of the PMF pollution event solutions to the exact time period range selected around the pollution event?

In our experience the solutions are robust once the time window is kept "short enough", so that the variability exhibited by the plume/episode forms a major part of the total variability in aerosol mass during the particular period. In this case the solutions are not sensitive to changes in the exact time window selection.

Extending the time window to longer periods, where the variability arising from other reasons (biogenic SV-OOA diurnal cycle, other, consecutive or partly overlapping pollution episodes from different sources, etc) starts to overly dominate the solution, does degrade the solutions quality. Typically this happens when extending the time window to several days (assuming a pollution episode of few hours). Ulbrich et al. (2009) estimate this variability "limit of detection" of separating a factor to be 5% of total variation in OA mass. We find their estimate agrees with our experience with the PMF runs of this work.

Did you consider as an alternative to this manual plume method to run a traditional PMF analysis on the entire dataset and identify plumes as time periods where the residuals of the PMF analysis are high?

The idea of looking at PMF residuals is an interesting one. We did not test it. However, it is a different philosophy in itself, as it pinpoints the mass/variation unexplained by the "standard" PMF model solution chosen. Some issues with this approach are:
As the unexplained mass/variation fundamentally does not equal a separate pollution (or even aerosol) type/source, but might equally well have to do with e.g. volatilization/condensation of semivolatiles, or, oxidation changing the composition of OA over time, or derive from a technical issue such as bad uncertainty estimate.
The definition of "traditional PMF" varies considerably and the analysis is usually at least as subjective as our current methodology, due to similar manual selection of correct solutions, but with fewer pointers to what would be the optimum solution.
Considering the close similarity of e.g. biogenic background SV-OOA and the sawmill SOA pollution, the two would likely get combined in one factor in "traditional PMF", producing only small residuals, and thus likely missing this important source. Likewise for the oxidized aerosol types (biogenic LV-OOA vs A-LV-OOA).
Residuals are additionally produced by violations of PMF's underlying assumptions, mainly the idea that factor mass spectral profiles are constant over time (e.g. Ulbrich et al., 2009) – this is often unrealistic for atmospheric aerosols, and the issue is exacerbated for long time series, presumably exhibiting more chemical processes.
To combat these effects were key reasons to split the PMF examination to small time windows in the first place. Using shorter timeframes model assumptions of constant factor profiles can be better assumed to hold, and the separation be more likely driven by actual source based separation than reflecting ongoing chemistry. In short, we foresee trying to pinpoint pollution episodes/types from the residuals would very likely run aground with even worse demarcation issues and subjectivity problems than the current OA mass based selection.

Discussion on these issues was added also to Sect. 2.3.1.

3) It is not clear to me why the pollution event PMF analysis used in this manuscript necessarily provides a more unambiguous separation of "pollution" and "background" than PMF analyses that are performed on the whole dataset. In fact, if the pollution event is simply a result of changes in wind direction that mixes in a different well mixed airmasses, then the PMF factor that is extracted would necessarily be just an average mass spectrum of all the sources present in the polluting airmass. No advantage would have been gained by this method to allow separation of individual sources and this would seem to be a weakness of this method. This aspect is not discussed in the manuscript.

This discussion also links to the earlier question and its answers.

If two sources of pollution are (practically) collocated, unconstrained PMF is unable to differentiate between the sources. This is an inherent limitation of the said data reduction method / receptor model (whether doing the entire data or an episode). In this case the sources are attributed to the same factor ("average mass spectrum").

However, assuming we are not only relying on one single source of a specific aerosol type, which we believe to be the case given e.g. the variable wind direction distributions (supplementary Figure S.11.), we would expect not all the emission sources will be similarly collocated (and emissions produced in in similar fractions), and thus some sources would produce a more "pure" sample of e.g. HOA or COA. This should be especially true for the close-by sources of e.g. passing cars or emissions from the forestry station. The collocated sources would then be expected show up between the pure samples, which is what we hypothesise to be happening with the weak (transported aerosols, A-SV-OOA) clusters.

We agree that this discussion on mixed pollution events is needed and have added it to the manuscript, to Sections 2.3.1, 3.4 and the conclusions.

Running PMF over the whole data set would not capture the plumes due to the 5% "limit of detection" mentioned earlier. Similarly, it would not provide the separated spectra needed for the clustering applied here. Finally, and our approach is also more robust when it comes to selecting optimal PMF solutions, as described in sections 2.2.2 and 2.3.1. We fix the rotation using "external" information on factor time-series, i.e. what a physically correct (albeit qualitative) description of time series behaviour is like.

4) The manuscript refers to ambiguities in PMF analysis as a weakness and implies that this analysis somehow solves or provides a better solution to this problem of ambiguity. In fact, the manuscript clearly states the difficulty of separating the various primary aerosol sources.

See previous answer.

Regarding the second part, the difficulty of separating primary sources spectra (from each other), discussed specifically in Sect. 3.4.2 is not connected to the question of rotational ambiguity of the PMF model, but the performance of distance metric used in clustering.

One of the advantages of the traditional method of doing PMF or ME- 2 on the entire dataset in this context could be the fact that it can exploit differences in temporal profiles of primary sources (i.e. different diurnal cycles) and also exploit the fact that source mass spectra are similar to allow for separation of multiple primary sources within a well mixed pollution event (i.e. a event such as that mentioned in comment 3 above).

As we can see from the Supplement Fig. S.11 and S.12, the pollution observations are very limited in number, and do not amount anywhere close to forming statistically relevant diurnal patterns

(partly also due to transport times). For a remote station such as SMEAR II, diurnality analysis of pollution plumes (whether by PMF or other receptor models), thus seems unfeasible.

Our intention is in no way to discredit PMF or constrained ME-2 itself, but merely to provide a robust, experimental basis for e.g. using objective and realistic constraints for an improved supervised analysis (e.g. constrained factors).

A comparison between the classification results and a traditional PMF of the entire dataset would have been a good way to address this and to highlight similarities and differences in results. The manuscript should more clearly state discuss the advantages/disadvantages of using this method compared to PMF.

A "traditional" (constrained) ME-2 analysis of two of the data sets ("March 2009" and "September 2008") has been published by Crippa et al. (2014), and is referenced in Sect 2.2.1. In their work Crippa et al. separated LV and SV OOA components, and additionally constrained BBOA and HOA using a reference spectrum (from Paris). In our work we compare our results against these reference spectra (described in Crippa et al., 2013), and find the similarities very high (Sect 3.4). A traditional PMF has been published by Corrigan et al., 2013, but it only managed to separate a BBOA factor besides the general SV and LV components. E.g. Canonaco et al., (2013) and Crippa et al., (2014), highlight the difficulties of separating primary sources for a rural background station like ours in an unconstrained analysis, undermining the feasibility of extracting a wide range of primary OA factors.

As suggested by referee 1, we have modified this section to focus on what scientific insights our approach offers in addition to PMF, rather than implying a "competition" between the two approaches.

5) The strongest part of this manuscript is the application and interpretation of the various clustering metrics to understand similarity and differences between the cluster spectra. It may be useful to highlight more strongly how these metrics could be applied to spectra obtained with typical PMF/ME-2 analysis. Would use of the cluster analysis metrics to reference spectra and PMF solutions provide a means of automating classification in PMF analysis? Also an intriguing part of this that could be discussed in more detail is the possibility to use the cluster analysis to define a-values and reference spectra for ME-2.

The prospect of automatic classification of PMF results is in our opinion definitely a feasible one, and indeed one of the motivations for this type of a study. The application could be, for example, in classifying a large set of bootstrapped PMF runs or a large number of PMF "seed" runs or Fpeak runs' results. Similarly, we hope the spectral similarity metric optimization would be useful for a) evaluation of PMF result factors' similarities against each other and b) identification of PMF result spectra (against library references, such as the AMS Spectral Database). We discuss the application of cluster centroid spectra and within-cluster variation as fitting input for a constrained ME-2 analysis (in the Introduction, Sect 3.6 and Conclusions), but have now highlighted this further, and added a paragraph in conclusions summarizing these future prospects. We thank the referee for these good suggestions, and have added emphasis to them.
* * *
**references:**

Canonaco, F., Crippa, M., Slowik, J., Baltensperger, U., and Prévôt, A.: SoFi, an IGOR-based interface for the efficient use of the generalized multilinear engine (ME-2) for the source apportionment: ME-2 application to aerosol mass spectrometer data, *Atmospheric Measurement Techniques*, 6, 3649-3661, 2013.

Crippa, M., DeCarlo, P., Slowik, J., Mohr, C., Heringa, M., Chirico, R., Poulain, L., Freutel, F., Sciare, J., and Cozic, J.: Wintertime aerosol chemical composition and source apportionment of the organic fraction in the metropolitan area of Paris, *Atmospheric Chemistry and Physics*, 13, 961-981, 2013.

Crippa, M., Canonaco, F., Lanz, V., Äijälä, M., Allan, J., Carbone, S., Capes, G., Ceburnis, D., Dall'Osto, M., and Day, D.: Organic aerosol components derived from 25 AMS data sets across Europe using a consistent ME-2 based source apportionment approach, *Atmospheric chemistry and physics*, 14, 6159-6176, 2014.

Corrigan, A., Russell, L., Takahama, S., Äijälä, M., Ehn, M., Junninen, H., Rinne, J., Petäjä, T., Kulmala, M., and Vogel, A.: Biogenic and biomass burning organic aerosol in a boreal forest at Hyytiälä, Finland, during HUMPPA-COPEC 2010, *Atmospheric Chemistry and Physics*, 13, 12233-12256, 2013.

Henry R.C.: Current factor analysis models are ill-posed. *Atmospheric Environment,* 21, 1815-1820, 1987.

Paatero, P.: Least squares formulation of robust non-negative factor analysis, *Chemometrics and intelligent laboratory systems*, 37, 23-35, 1997.

Paatero, P., Hopke, P. K., Song, X.-H., and Ramadan, Z.: Understanding and controlling rotations in factor analytic models, *Chemometrics and intelligent laboratory systems*, 60, 253-264, 2002.

Paatero, P., Eberly, S., Brown, S. G., and Norris, G. A.: Methods for estimating uncertainty in factor analytic solutions, *Atmos. Meas. Tech.*, 7, 781-797, 2014.

Ulbrich, I., Canagaratna, M., Zhang, Q., Worsnop, D., and Jimenez, J.: Interpretation of organic components from Positive Matrix Factorization of aerosol mass spectrometric data, *Atmospheric Chemistry and Physics*, 9, 2891-2918, 2009.